JGP | Journal of General Physiology

# Transport of metformin metabolites by guanidinium exporters of the small multidrug resistance family

Rachael M. Lucero[1], Kemal Demirer[2], Trevor Justin Yeh[3], and Randy B. Stockbridge[1,2,3]

**Proteins from the small multidrug resistance (SMR) family are frequently associated with horizontally transferred multidrug resistance gene arrays found in bacteria from wastewater and the human-adjacent biosphere. Recent studies suggest that a subset of SMR transporters might participate in the metabolism of the common pharmaceutical metformin by bacterial consortia. Here, we show that both genomic and plasmid-associated transporters of the $SMR_{Gdx}$ functional subtype export byproducts of microbial metformin metabolism, with particularly high export efficiency for guanylurea. We use solid-supported membrane electrophysiology to evaluate the transport kinetics for guanylurea and native substrate guanidinium by four representative $SMR_{Gdx}$ homologs. Using an internal reference to normalize independent electrophysiology experiments, we show that transport rates are comparable for genomic and plasmid-associated $SMR_{Gdx}$ homologs, and using a proteoliposome-based transport assay, we show that 2 proton:1 substrate transport stoichiometry is maintained. Additional characterization of guanidinium and guanylurea export properties focuses on the structurally characterized homolog, Gdx-Clo, for which we examined the pH dependence and thermodynamics of substrate binding and solved an x-ray crystal structure with guanylurea bound. Together, these experiments contribute in two main ways. By providing the first detailed kinetic examination of the structurally characterized $SMR_{Gdx}$ homolog Gdx-Clo, they provide a functional framework that will inform future mechanistic studies of this model transport protein. Second, this study casts light on a potential role for $SMR_{Gdx}$ transporters in microbial handling of metformin and its microbial metabolic byproducts, providing insight into how native transport physiologies are co-opted to contend with new selective pressures.**

## Introduction

Membrane transporters are essential for microbial survival in dynamic environments. They bridge the interior of the cell with the external environment and permit the translocation of nutrients, metabolic byproducts, and toxins across the membrane barrier. In particular, efflux pumps are a first line of defense against a variety of xenobiotics, including anthropogenic chemicals (Kim et al., 2021; Paulsen, 2003). One reflection of the fitness advantage provided by these exporters is their frequent association with horizontal gene transfer (HGT) elements such as integron/integrase sequences and plasmids, which permit useful genes to be shared among bacterial populations. HGT-associated genes encoding drug exporters are especially common among isolates from hospitals, wastewater, agriculture, and other human-adjacent contexts (Pal et al., 2015).

Representatives of the small multidrug resistance (SMR) family of proton-coupled antiporters are among the most common HGT-associated exporters (Pal et al., 2015). These ∼100 residue proteins possess four transmembrane helices per monomer and assemble as antiparallel dimers (Fleishman et al., 2006; Kermani et al., 2020, 2022). Structures of representative SMRs show a deep aqueous substrate binding pocket with a critical pair of glutamate residues at the bottom (Kermani et al., 2020, 2022). Substrate and protons compete for the binding of these glutamates, ensuring the alternating occupancy inherent to antiport mechanisms (Muth and Schuldiner, 2000). Two SMR subtypes with distinct substrate specificities are commonly associated with HGT (Burata et al., 2022; Kermani et al., 2018; Slipski et al., 2020). These are termed $SMR_{Gdx}$ (guanidinium export) and $SMR_{Qac}$ (quaternary ammonium cation). The $SMR_{Qac}$ proteins are promiscuous exporters of polyaromatic and quaternary ammonium antimicrobials, including common household and hospital antiseptics such as benzalkonium (Saleh et al., 2018; Yerushalmi et al., 1995). Quaternary ammonium antiseptics are one of the original modern antimicrobials, commonly used since the 1930s. The $SMR_{Qac}$s are perhaps the first, and remain among the most common, HGT-associated efflux pumps (Gillings et al., 2008; Zhu et al., 2017). In contrast, the rationale for the widespread association between HGT elements and the $SMR_{Gdx}$ is not as obvious. In their major physiological context, $SMR_{Gdx}$ export the nitrogenous waste product guanidinium ($Gdm^+$; Kermani

---

[1]Program in Chemical Biology, University of Michigan, Ann Arbor, MI, USA;   [2]Department of Molecular, Cellular, and Developmental Biology, University of Michigan, Ann Arbor, MI, USA;   [3]Program in Biophysics, University of Michigan, Ann Arbor, MI, USA.

Correspondence to Randy B. Stockbridge: stockbr@umich.edu.

et al., 2018; Nelson et al., 2017), a compound that is widespread in microbial metabolism (Breaker et al., 2017; Funck et al., 2022; Schneider et al., 2020; Sinn et al., 2021; Wang et al., 2021). The SMR$_{Gdx}$ do not provide robust resistance to classical antimicrobials or antiseptics (Chung and Saier, 2002; Kermani et al., 2018). However, an emerging body of literature suggests that even pharmaceuticals that are not used explicitly as antimicrobials also shape bacterial communities in the human microbiome and other human-associated environments (Maier et al., 2018).

One such pharmaceutical is the biguanide antidiabetic metformin. The most frequently prescribed drug worldwide, over 150 million patients are prescribed metformin annually to manage type II diabetes (Lunger et al., 2017). Metformin is typically dosed in gram quantities daily and is excreted in an unaltered form (Gong et al., 2012; Corcoran and Jacobs, 2022). Metformin and its associated degradation product guanylurea are the most prevalent anthropogenic chemicals in wastewater globally. Concentrations have been measured up to the low µM range in sampled waste and surface waters, and these compounds are not removed through typical wastewater treatment protocols (Briones et al., 2016; Golovko et al., 2021). As a result, these compounds have accumulated to levels of environmental concern in surface water worldwide (Balakrishnan et al., 2022; Briones et al., 2016; Elizalde-Velazquez and Gomez-Olivan, 2020; Scheurer et al., 2012). Metformin is also associated with changes in the composition of microbial communities including the gut microbiome (Vich Vila et al., 2020; Wu et al., 2017) and in wastewater treatment plants (Briones et al., 2016). In some cases, metformin may act as a co-selective agent, enhancing the survival of antibiotic-resistant bacteria in the presence of antibiotics (Wei et al., 2022). However, other recent studies have isolated bacteria that utilize metformin as a nitrogen and/or carbon source (Chaignaud et al., 2022; Li et al., 2023; Martinez-Vaz et al., 2022), suggesting that biodegradation of metformin and guanylurea may be a viable strategy for remediation of these compounds.

Studies on metformin degradation by microbial communities suggest that SMR transporters might have an emerging role in metformin biodegradation. For example, two identical, adjacent open reading frames encoding an SMR$_{Gdx}$ protein were identified on the same plasmid as other genes that contribute to metformin degradation by a wastewater treatment plant isolate (Martinez-Vaz et al., 2022). We previously showed that this protein possesses guanylurea transport activity (Martinez-Vaz et al., 2022). In an independent study, a transcriptional analysis of a metformin-degrading *Aminobacter* strain showed a 30-fold increase in gene expression of an SMR$_{Gdx}$ transporter in metformin-grown cells (Li et al., 2023). On the basis of these studies, pathways for the full breakdown of metformin by bacterial consortia have been proposed. In such pathways, SMR$_{Gdx}$ transporters would provide a key step in the process, export of the intermediate guanylurea (Fig. 1 A).

In this paper, we investigate whether several genomic- and plasmid-associated SMRs (Fig. 1 B and Table S1) transport metformin or other byproducts of microbial metformin metabolism. For our initial screen, we examined four SMR$_{Gdx}$ homologs and two SMR$_{Qac}$ homologs. The SMR$_{Gdx}$ homologs we examined include (1) the structurally characterized genomic protein from *Clostridales* oral taxon 876, Gdx-Clo (Kermani et al., 2020); (2) the genomic *Escherichia coli* homolog Gdx-Eco (Kermani et al., 2018); (3) a common plasmid-borne variant isolated from multiple species of γ-proteobacteria, Gdx-pPro (Slipski et al., 2020), which shares 81% sequence identity with Gdx-Eco; and (4) a plasmid-borne variant isolated from *Aminobacter* sp. MET, which uses metformin as a sole nitrogen source, Gdx-pAmi (Martinez-Vaz et al., 2022). We also selected two representatives of the SMR$_{Qac}$ subtype; exemplar EmrE from *E. coli* and QacE, the most common integron- and plasmid-associated sequence (Burata et al., 2022). We show that efficient guanylurea transport is a general property of the SMR$_{Gdx}$ subtype, but not of SMR$_{Qac}$, and that other metformin degradation products are also transported by SMR$_{Gdx}$. We characterize the transport kinetics and proton-coupling stoichiometry of a representative plasmid-borne and genomic SMR$_{Gdx}$ and determine the structure of a representative SMR$_{Gdx}$ with guanylurea bound. This work provides a case study into bacterial co-option of existing metabolic transporters to deal with novel xenobiotics. Furthermore, this study provides the foundational biochemical characterization of the SMR$_{Gdx}$ subtype, which will support future efforts to understand detailed molecular mechanisms of substrate transport by this family of proteins.

## Materials and methods
### Phylogeny preparation
SMR sequences from representative genomes and from Integrall (Moura et al., 2009), a database of integron-associated genes, were aligned using MUSCLE (Edgar, 2004). A phylogeny was constructed using PhyMl3.0 (Guindon et al., 2010) and visualized using FigTree (http://tree.bio.ed.ac.uk/software/figtree).

### Transporter expression, purification, and reconstitution
Gdx-Clo (Kermani et al., 2018), Gdx-Eco (Kermani et al., 2018), EmrE (Kermani et al., 2022), and Gdx-pAmi (Martinez-Vaz et al., 2022) construct design and purification have been described previously. For QacE and Gdx-pPro, synthetic gene-blocks (Integrated DNA Technologies) were cloned into a pET21b vector with an N-terminal hexahistidine tag and LysC and thrombin recognition sequences. Proteins were overexpressed in C41(DE3). Expression was induced by the addition of 0.2 mM IPTG for 3 h. Cells were lysed and extracted with 2% n-decyl-β-D-maltoside (DM) for 2 h. After pelleting insoluble cell debris, proteins were purified using cobalt affinity resin. Wash buffer contained 25 mM Tris, pH 8.5, 150 mM NaCl, and 5 mM DM. For Gdx-pAmi, NaCl concentration was increased to 500 mM NaCl. The affinity column was washed with wash buffer, then wash buffer with 10 mM imidazole, prior to elution with wash buffer with 400 mM imidazole. For Gdx-Clo and Gdx-Eco, histidine tags were cleaved with LysC (200 ng/mg of protein; 2 h at room temperature; New England Biolabs), and for all others, histidine tags were cleaved with thrombin (1 U/mg of protein, overnight at room temperature; MilliporeSigma). Proteins were further purified using a gel filtration Superdex200 column (Cytiva) equilibrated with 100 mM NaCl, 10 mM N-2-hydroxyethylpiperazine-N′-2-ethanesulfonic acid (HEPES), pH 7.5, and

5 mM DM. Purified proteins were stored at 4°C for up to 5 days before detergent binding assays. To prepare proteoliposomes for electrophysiology assays, purified protein was mixed with *E. coli* polar lipid extract (10 mg/ml; Avanti Polar Lipids) solubilized with 35 mM 3-[(3-cholamidopropyl)dimethylammonio]-1-propanesulfonate (CHAPS) at a protein to lipid ratio of 40 µg SMR transporter: mg lipid (1:370 protein:lipid molar ratio) prior to detergent removal by dialysis. For preparations that included Fluc-Bpe, liposomes were reconstituted with a molar ratio of 0.3 Fluc-Bpe:1 SMR$_{Gdx}$: 5920 lipid (1 µg Fluc-Bpe and ~2.5 µg SMR$_{Gdx}$ per mg lipid). For liposome transport assays, proteoliposomes were prepared similarly, except that a 2:1 mixture of 1-palmitoyl, 2-oleoylphosphatidylethanolamine (POPE) and 1-palmitoyl, 2-oleoylphosphatidylglycerol (POPG) (10 mg/ml; Avanti Polar Lipids) was used with 0.2 µg protein/mg lipid. Proteoliposomes were stored at −80°C until use.

### Solid-supported membrane (SSM) electrophysiology

SSM experiments were performed using SURFE$^2$R N1 instrument (Nanion Technologies). Sensors were prepared with a 1,2-diphytanoyl-sn-glycero-3-phosphocholine (DPhPC) lipid monolayer according to published protocols (Bazzone et al., 2017). Each sensor's capacitance and conductance were verified before use (<80 nF capacitance, <50 nS conductance) using Nanion software protocols. Proteoliposome stock was diluted 1:25 in assay buffer (100 mM KCl, 100 mM KPO$_4$, pH 7.5) prior to adsorption to the DPhPC monolayer. For substrate screening experiments, reference substrate samples (Gdm$^+$ for SMR$_{Gdx}$, and TPA$^+$ for SMR$_{Qac}$) were checked periodically to test for the stability of the sensor; if the current amplitude of the reference compound differed by >10% on one sensor, it indicated the desorption of liposomes, and the sensor was not used for further experiments. For every SSM electrophysiology experiment, we used the same general solution exchange protocol: after 2 s perfusion with a non-activating solution, we perfused substrate-containing buffer for 2 s and then returned to the equilibrium condition with a 4 s perfusion of non-activating buffer. An example of a full experiment is shown in Fig. S1. For experiments with the S104C/A70C crosslinked variant, reducing conditions were established by the addition of 2.5 mM tris(2-carboxyethyl) phosphine (TCEP) to the activating and non-activating buffers.

### Radioactive Gdm$^+$ exchange assay

This assay was performed exactly as described previously (Kermani et al., 2018). Briefly, proteoliposomes were prepared with 10 mM internal Gdm$^+$. 20 µM external $^{14}$C-labeled guanidinium (American Radiolabelled Chemicals, Inc.) was added to initiate the exchange reaction and quenched at timepoints by passing the sample over a cation exchange column. Internalized radiolabeled Gdm$^+$ was measured by scintillation counting. Reducing conditions were established by the inclusion of 2.5 mM tris(2-carboxyethyl)phosphine (TCEP) in the buffers.

### Quantitative Western blot

For experiments to assess coreconstitution efficiency, Fluc proteins bore an MPER epitope tag, which does not alter channel function or biochemistry (McIlwain et al., 2021a), and the

SMR$_{Gdx}$ homolog bore a histidine tag. To detect MPER epitope tags, we used the primary antibody VRC42 (Krebs et al., 2019), and to detect the histidine tags, the primary antibody was from Genscript. Quantification was performed using ImageJ (Schneider et al., 2012).

### Pyranine stoichiometry assay

Proteoliposomes were reconstituted with an internal buffer of 25 mM HEPES, pH 7.53, 100 mM NaCl, 100 mM KCl, and preloaded with 0.4 mM substrate (Gdm$^+$ or guanylurea) and 1 mM pyranine (trisodium 8-hydroxypyrene-1, 3, 6-trisulfonate; Sigma-Aldrich) using three freeze/thaw cycles. Unilamellar liposomes were formed by extrusion through a 400-nm membrane filter and the external pyranine was removed by passing liposomes through a Sephadex G-50 column spin column equilibrated in an internal buffer with substrate. The external assay buffers contained 25 mM HEPES, pH 7.53, 0.4 mM substrate, and varying KCl concentration (3–46 mM) to establish the membrane potential, with NaCl to bring the total salt concentration to 200 mM. Proteoliposomes were diluted 200-fold into the external buffer, and after ~30 s to establish a baseline, valinomycin (final concentration 0.2 ng/ml) was added together with the substrate (final concentration 4 mM). Fluorescence spectra were monitored ($\lambda_{ex}$ = 455 nm; $\lambda_{em}$ = 515 nm) for ~300 s. The membrane potential was calculated using the Nernst potential for K$^+$:

$$\psi_{calc} = \frac{RT}{F} \ln \frac{[K^+]_{out}}{[K^+]_{in}}. \qquad (1)$$

Fluorescence emission time courses were corrected for baseline drift measured prior to substrate and valinomycin addition. The stoichiometry was determined from the voltage at which electrochemical equilibrium occurred (no change in fluorescence over time) using the following equation:

$$E_{rev} = \left( \frac{n}{m-n} * \frac{RT}{F} \ln \frac{[substrate^+]_{in}}{[substrate^+]_{out}} \right), \qquad (2)$$

where $n$ and $m$ represent the stoichiometric coefficients of substrate and protons, respectively.

### Tryptophan fluorescence

Fluorescence emission spectra ($\lambda_{ex}$ = 280 nm, $\lambda_{ex}$ = 300–400 nm) were collected for 1 µM purified protein in assay buffer containing 200 mM NaCl, 10 mM HEPES, 10 mM bicine, 10 mM NaPO$_4$, and 5 mM DM, with pH adjusted from 6.5 to 9.0. Substrate was added from a stock solution prepared in assay buffer. For Gdm$^+$ titrations, the change in fluorescence, $F$, as a function of substrate fit to a single site binding isotherm,

$$\Delta F = \left( \frac{F_{max}[S]}{K_d + [S]} \right). \qquad (3)$$

For guanylurea titration, binding data fit to a single site binding isotherm with correction for a linear, non-specific binding component, $c$:

$$\Delta F = \left( \frac{F_{max}[S]}{K_d + [S]} \right) + c[S]. \qquad (4)$$

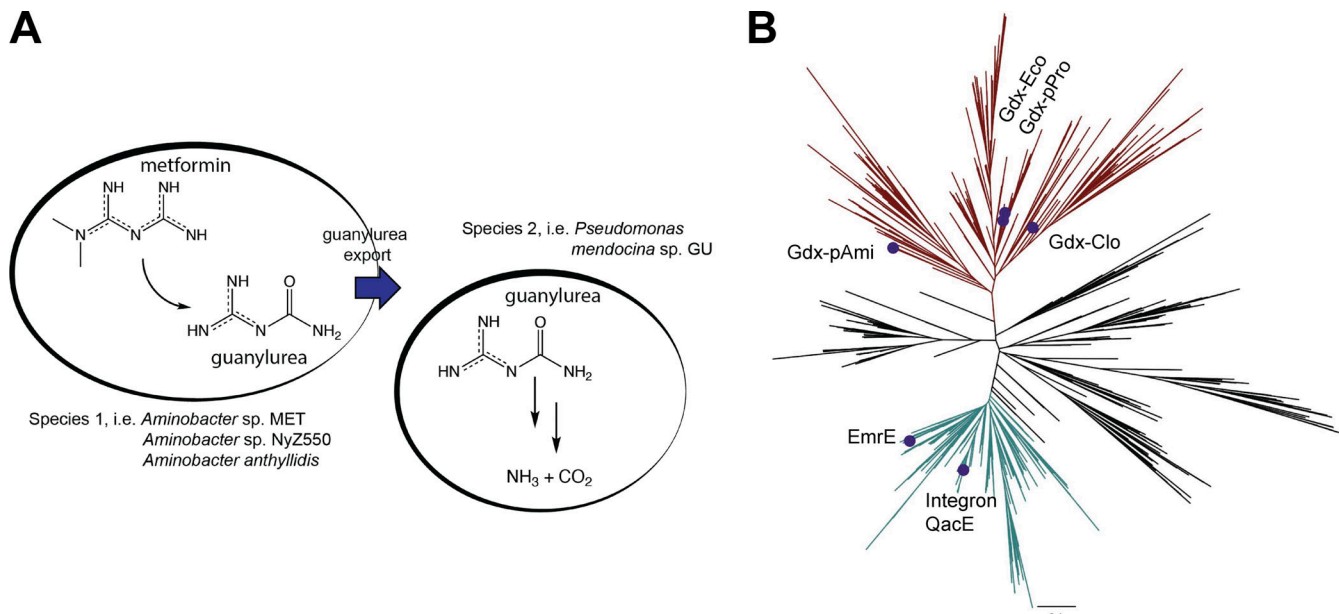

**Figure 1.** **SMR physiology and phylogenetic distribution. (A)** Schematic showing hypothesized role for horizontally transferred SMR_Gdx homologs in biodegradation of metformin by bacterial consortia. Species and degradation pathways are described in Chaignaud et al. (2022); Li et al. (2023); Martinez-Vaz et al. (2022); and Tassoulas et al. (2021). **(B)** Phylogeny of the SMR family. SMR_Gdx is shown in rust and SMR_Qac in teal. Proteins examined in this study are indicated.

To derive the $K_a$ values and $K_d$ values from the apparent $K_d$ measured as a function of pH, we used the following equation, which uses the approximation that the protonatable E13 sidechains have equal $K_a$ values:

$$K_{d,app} = (K_d) * \left(1 + \left(\frac{[H^+]}{K_a}\right)\right) * \left(1 + \left(\frac{[H^+]}{K_a}\right)\right). \quad (5)$$

### Isothermal titration calorimetry (ITC)

ITC experiments were conducted using a low-volume Nano ITC instrument (TA Instruments). Freshly purified protein (650 µM) in 10 mM 4-(2-hydroxyethyl)-1-piperazinepropanesulfonic acid (EPPS), pH 8.53, 100 mM NaCl, and 4 mM DM was titrated with 20 mM Gdm⁺ or 10 mM guanylurea prepared in the same buffer. For each experiment, 300 µl of 700 µM Gdx-Clo was loaded in the sample chamber maintained at 25°C with 350 rpm stirring speed. The injection syringe contained 500 µl of buffer-matched substrate (20 mM Gdm⁺ or 10 mM guanylurea). The sample was titrated (0.75 µl injections) at 100-s increments. Once an acceptable baseline slope was achieved (0.30 µW/h and 0.03 µW standard deviation), a 200 s baseline (~112 µW) was taken prior to beginning titrations. Data were analyzed using NanoAnalyze software.

### Structure of Gdx-Clo in complex with guanylurea

The crystallization chaperone monobody L10 was prepared as described previously (Kermani et al., 2020, 2022). Freshly purified Gdx-Clo (10 mg/ml) and L10 monobody (10 mg/ml, supplemented with 4 mM DM) were mixed at a 1:1 ratio. Guanylurea and lauryldimethylamine-N-oxide (LDAO; Anatrace) were added to a final concentration of 10 and 6.6 mM, respectively, and combined in a 1:1 ratio with crystallization solution. Crystals formed at room temperature after ~7 days in 0.1 M HEPES, pH 7.0, 0.1 M calcium acetate, and 31% PEG600. Data were collected

at the Life Sciences Collaborative Access Team at the Advanced Photon Source, Argonne National Laboratory. Data were processed using DIALS (Winter et al., 2018) software and subjected to anisotropic truncation using Staraniso (Tickle et al., 2018). Phaser (McCoy et al., 2007) was used for molecular replacement with Gdx-Clo and L10 monobodies (PDB ID 6WK9) as search models. Coot (Emsley et al., 2010) and Phenix (Liebschner et al., 2019) were used for iterative rounds of model building and refinement.

### Online supplemental material

Fig. S1 shows the solution exchange protocol and example trace for SSM electrophysiology experiments described in this manuscript. Fig. S2 shows size exclusion chromatograms for six proteins in this study. Fig. S3 shows representative current traces for substrates and transporter data summarized in Fig. 1, and no-protein controls. Fig. S4 shows representative current traces for Gdm⁺ and guanylurea titrations of protein-free liposomes. Fig. S5 shows representative current traces for guanylurea perfusion of Fluc-Bpe and fluoride perfusions of Gdx-Eco, Gdx-pPro, and Gdx-pAmi. Fig. S6 shows tryptophan fluorescence spectra and fits to binding isotherms for all data reported in Fig. 6 and Table 2. Table S1 shows coding sequences for transporters examined in this study. Table S2 shows reconstitution efficiencies of SMR_Gdx homologs assessed by quantitative Western blot.

## Results

### Guanylurea transport is general among SMR_Gdx homologs

We first sought to determine whether transport of guanylurea is widespread among SMR homologs, and whether other metformin metabolites might also be exported by transporters from this family. We selected several SMRs that could be purified with monodispersed size exclusion chromatograms (Fig. S2),

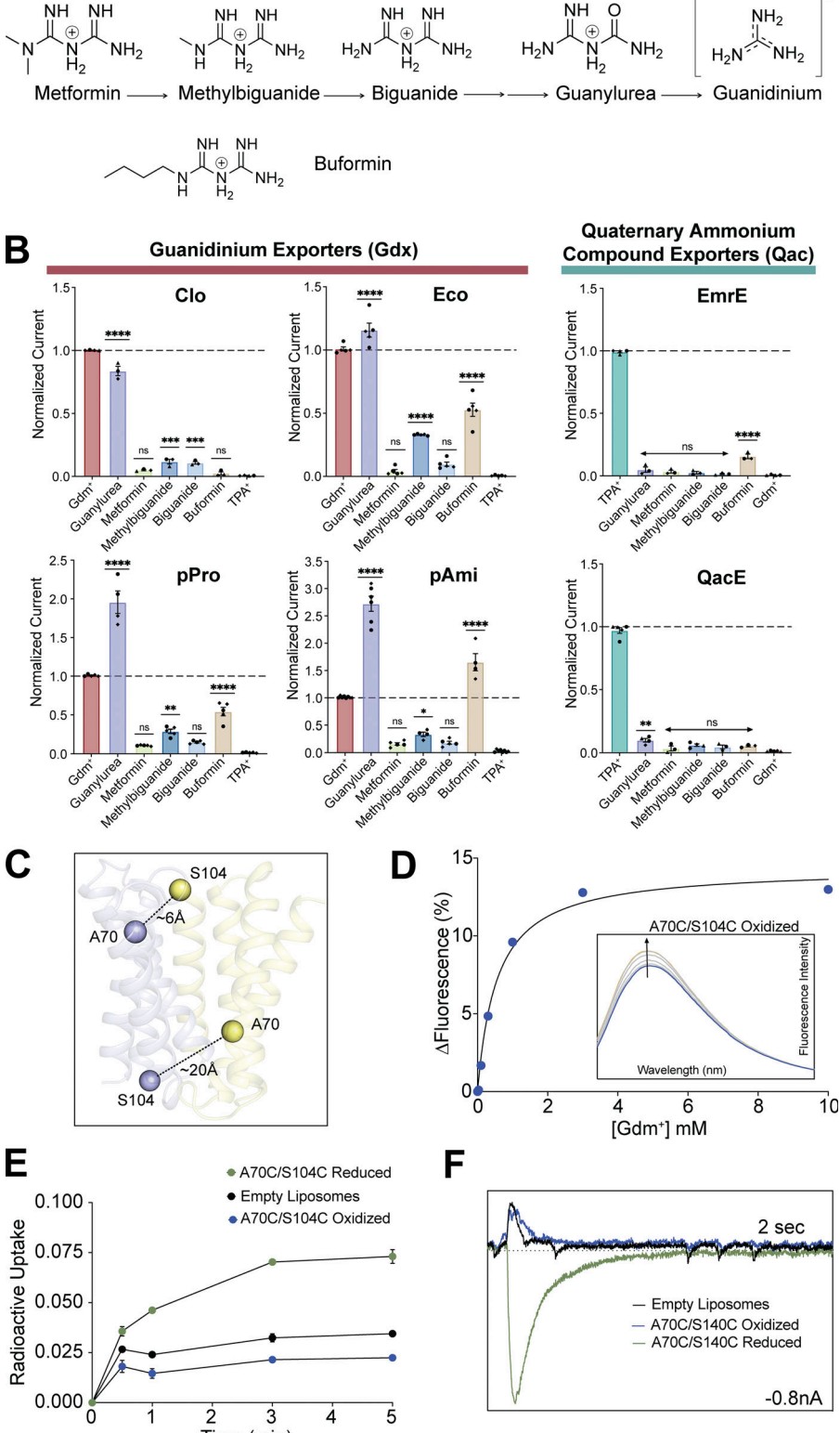

**Figure 2. Screen for transport of metformin metabolites by SMR homologs. (A)** Chemical structures of metformin metabolites and metformin analog buformin. **(B)** Amplitude of transport currents evoked by perfusion with 2 mM substrate. Current amplitudes are normalized to a positive control (Gdm⁺ for SMR_Gdx and TPA⁺ for SMR_Qac) collected on the same sensor. Each datapoint represents a measurement from a single independent sensor. Sensors were prepared from at least two independent biochemical purifications; each biochemical preparation is represented by a different shaped point. The bars show the mean and SEM of measurements from different sensors. P values were calculated for comparisons with the negative control samples (TPA⁺ for SMR_Gdx and Gdm⁺ for SMR_Qac) using one-way ANOVA. Significance is not calculated for the positive control samples used for normalization (Gdm⁺ for SMR_Gdx and TPA⁺ for SMR_Qac). **(C)** Structure of Gdx-Clo with locations of and approximate distances between the A70C and S104C mutations shown. Dimer subunits are in yellow and blue. **(D)** Gdm⁺ binding to A70C/S104C under oxidizing conditions was measured using tryptophan fluorescence. Inset: Arrow represents increase in fluorescence peak upon Gdm⁺ titration. Points and error bars represent the mean and SEM of three independent replicates. The solid line represents a fit to a single-site binding model with a $K_d$ value of 560 µM. **(E)** Timecourse of radiolabeled Gdm⁺ exchange into liposomes with Gdx-Clo A70C/S104C under oxidizing (blue) and reducing (green) conditions. A no-protein control (black) is shown for comparison. Points and error bars represent the mean and SEM of three replicate measurements. **(F)** Representative SSM electrophysiology traces were elicited by perfusion of Gdx-Clo A70C/S104C with 1 mM Gdm⁺ under oxidizing (blue) or reducing (green) conditions. No protein-control is shown in black. The box edges are 2 s and 0.8 nA, respectively.

including both genomic- and plasmid-associated SMR_Qac and SMR_Gdx representatives (see Fig. 1). We screened a series of metformin metabolites for transport using SSM electrophysiology (Fig. 2, A and B). For these experiments, purified proteins are reconstituted into proteoliposomes, which are then capacitively coupled to an electrode to monitor charge movement across the liposome membrane (Bazzone et al., 2017). Because of their antiparallel topology, homodimeric SMR transporters possess twofold symmetry with identical inward- and outward-facing structures (Morrison et al., 2011); thus, in

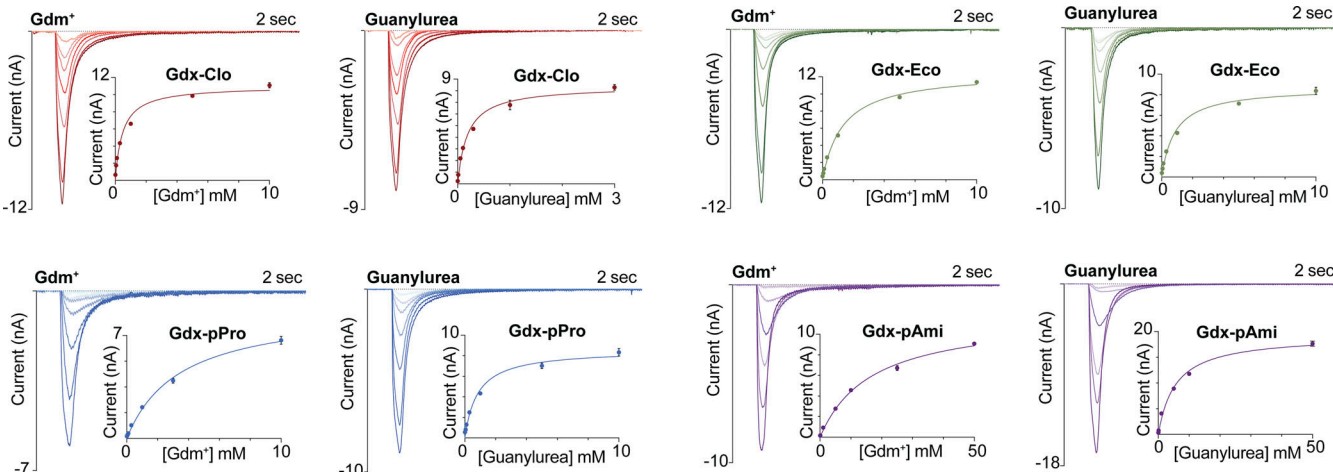

Figure 3. **Maximal current amplitudes as a function of Gdm+ or guanylurea concentration.** Representative transport currents for concentration series of the indicated substrates. Inset: Maximum current amplitude as a function of substrate concentration. Solid line represents a fit to the Michaelis–Menten equation. Each of these representative plots was obtained on a single sensor, and error bars represent the SEM for triplicate measurements on that single sensor. $K_m$ values reported in Table 1 represent averages from at least three independent sensors prepared from two to three independent protein preps.

contrast to most transporters, orienting the proteins in the reconstituted liposome system is not necessary. All compounds were tested for transport at 2 mM, and for each substrate, we confirmed that protein-free liposomes did not exhibit pronounced currents (Fig. S3). Since the efficiency of proteoliposome adsorption to the sensors' monolayer is variable, we included a positive control compound to benchmark the currents for test substrates evaluated on the same sensor: Gdm+ for SMR$_{Gdx}$ proteins, and tetrapropylammonium (TPA+) for SMR$_{Qac}$ proteins.

For all SMR$_{Gdx}$ homologs, we observed negative capacitive currents for both Gdm+ and guanylurea, consistent with electrogenic proton-coupled substrate antiport (Fig. S3). The best characterized SMR$_{Gdx}$ homolog, Gdx-Clo, transported only Gdm+ and guanylurea. However, the other three SMR$_{Gdx}$ homologs tested also transported singly substituted biguanides, including the metformin degradation product methylbiguanide and the related antidiabetic drug buformin. Metformin, a doubly substituted biguanide, exhibited currents barely above the detectable limit by SMR$_{Gdx}$ proteins. These observations are congruent with prior observations that guanidinium ions with single hydrophobic substitutions are transported by SMR$_{Gdx}$, but that doubly substituted guanidiniums are not (Kermani et al., 2020). The SMR$_{Qacs}$ examined, EmrE and integron-associated

QacE, did not exhibit transport currents for this series of compounds.

For all SSM electrophysiology experiments, the shapes of the substrate-induced currents are characteristic of transport, rather than electrogenic presteady binding events. However, we sought to confirm this interpretation for at least one transporter/substrate pair. Using the structure of Gdx-Clo (Kermani et al., 2020), we introduced a pair of cysteines, A70C and S104C, that are within the crosslinking distance (~6 Å) on the open side of the transporter, but that increase in distance when the transporter changes conformation. We expected that the formation of a crosslink would lock the transporter in one open conformation, impairing transport, with little effect on substrate binding (Fig. 2 C). Indeed, under oxidizing conditions, the $K_d$ for Gdm+ binding is within a factor of two of WT (Nelson et al., 2017), but radioactive Gdm+ exchange is greatly reduced to near-background levels (Fig. 2, D and E). Substrate exchange is restored in reducing conditions. SSM electrophysiology recapitulates this observation: under oxidizing conditions, the SSM electrophysiology traces of Gdx-Clo A70C/S104C are indistinguishable from those of protein-free liposomes, but the inclusion of a reducing agent elicits characteristic transport currents (Fig. 2 F). Thus, although we have evidence that Gdx-Clo A70C/S104C binds substrate normally when locked, we do not see any evidence of pre-steady state binding currents.

### Kinetics and proton coupling for Gdm+ and guanylurea transport

To compare the kinetic properties for the transport of guanylurea and the physiological substrate Gdm+, we measured peak amplitudes of the capacitive currents for the four SMR$_{Gdx}$ homologs as a function of substrate concentration. Assuming that currents reflect steady-state transport and not pre-steady-state binding events (as confirmed for Gdx-Clo/Gdm+ in Fig. 2 F), the current amplitudes reflect the initial rate of transport (Bazzone et al., 2017, 2022, 2023), and their concentration dependence

Table 1.   **$K_m$ values determined using SSM electrophysiology (pH 7.5)**

|  | Gdm+ (mM) ± SEM | Guanylurea (mM) ± SEM |
|---|---|---|
| Gdx-Clo | 0.5 ± 0.1 | 0.22 ± 0.06 |
| Gdx-Eco | 1.7 ± 0.5 | 0.85 ± 0.1 |
| Gdx-pPro | 2.9 ± 0.5 | 0.9 ± 0.2 |
| Gdx-pAmi | 15.7 ± 3.3 | 5.2 ± 2.0 |

Values represent mean and SEM from at least three independent sensors prepared from two to three independent protein preps.

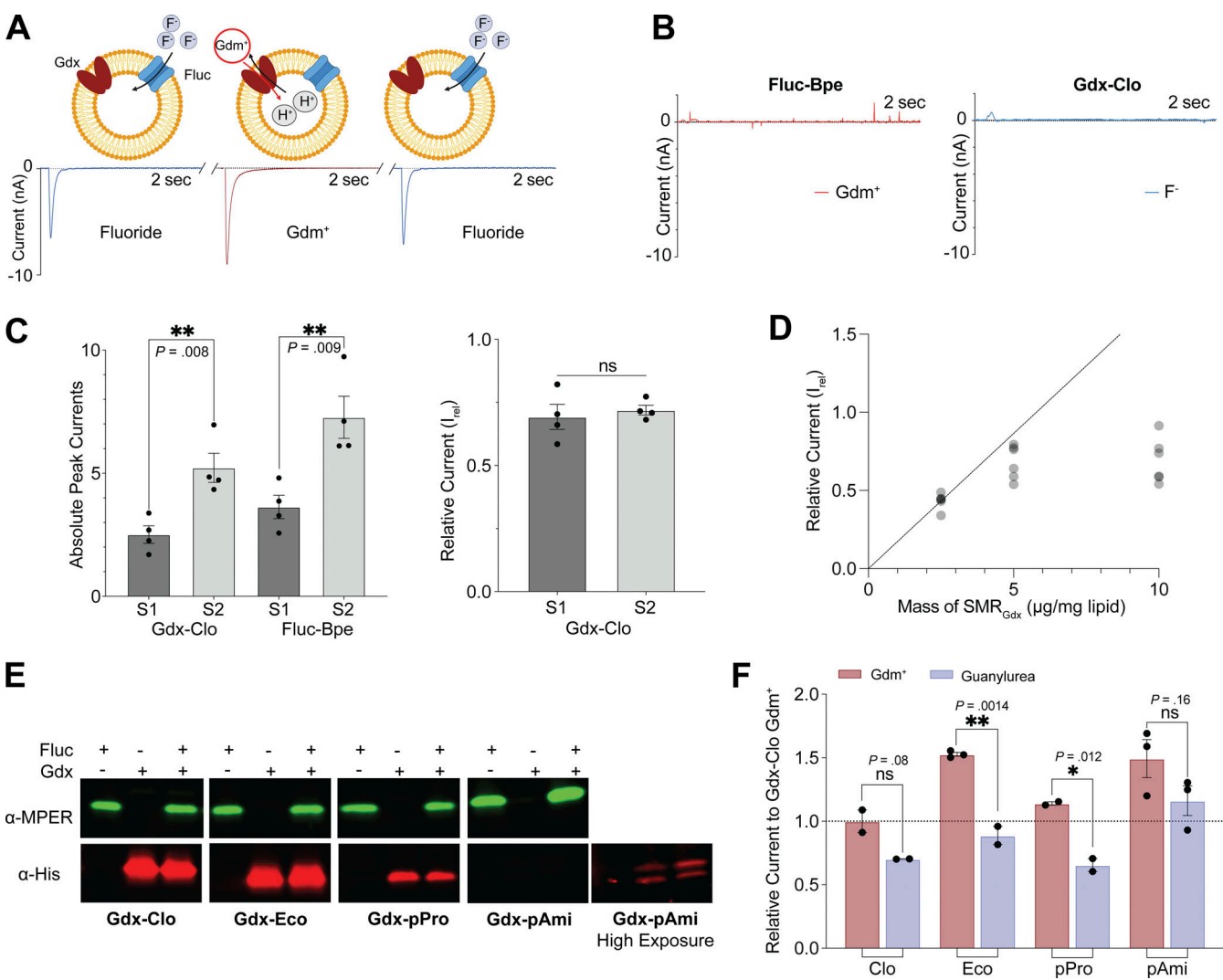

Figure 4. **Comparison of maximal velocities for different SMR homologs using an internal reference. (A)** Schematic showing experimental strategy of coreconstitution of target transporter and fluoride channel Fluc-Bpe with alternating substrate perfusions. The breaks in the trace represent perfusion with non-activating buffer to return to the equilibrium condition. Cartoon made using Biorender. **(B)** Current traces for Gdx-Clo and Fluc-Bpe reconstituted individually do not show substrate cross-reactivity. Tests for cross-reactivity by guanylurea and other SMR$_{Gdx}$ homologs are shown in Fig. S4. **(C)** Left, peak current amplitude for Gdm$^+$ and fluoride currents for two examples of independent sensor preparations. Right: Relative Gdm$^+$/fluoride current amplitude (I$_{rel}$) for the sensors shown in the left panel. Error bars represent the SEM of individual replicates shown as points. P values are shown (two-tail $t$ test). **(D)** I$_{rel}$ as a function of increasing SMR$_{Gdx}$ (Bpe-Fluc held constant at 1 µg/mg lipid). The dashed line represents expected peak current amplitude for a linear response. **(E)** Western blot analysis of proteoliposomes reconstituted with MPER-tagged Fluc-Bpe (green) and His-tagged Gdx-Clo (red) individually or together. Full membrane images in source data. The reconstitution efficiency of Gdx-pAmi was lower than for the other proteins, so we also collected an image at higher exposure for visualization purposes. Quantification (all at the same exposure time) is reported in Table S2. **(F)** Currents for Gdm$^+$ and guanylurea transport by four SMR$_{Gdx}$ homologs normalized against internal Fluc-Bpe reference currents. Each substrate was perfused at a concentration fivefold higher than the K$_m$ values measured in Fig. 3 to compare maximal turnover velocities among the different transporters. Error bars represent the SEM of individual replicates from different sensors shown as points. Significance calculations were performed using two-tailed $t$ test. Source data are available for this figure: SourceData F4.

follows Michaelis–Menten kinetics (Fig. 3 and Table 1). For all four homologs, the K$_m$ value for guanylurea was approximately twofold lower than that of Gdm$^+$. However, the absolute K$_m$ values varied over a factor of ~50 among these proteins. The genomic Gdx-Clo exhibited the lowest K$_m$ values (500 µM for Gdm$^+$ and 220 µM for guanylurea), and the plasmid-associated Gdx-pAmi exhibited the highest K$_m$ values (16 mM for Gdm$^+$ and 5 mM for guanylurea). We confirmed that protein-free liposomes do not exhibit negative capacitive currents characteristic of transport; at the highest substrate concentrations, we

observe small positive currents, indicative of interactions with the membrane (Fig. S4).

Our experiments thus far do not allow a comparison of transport rates among SMR homologs. The adsorption of proteoliposomes to the sensor is subject to considerable variability from experiment to experiment, so the current measurements from different sensors cannot be quantitatively compared (Barthmes et al., 2016). Adsorption efficiency can vary from day to day, by experimenter or by sensor batch. To normalize maximal currents obtained on different sensors and thus

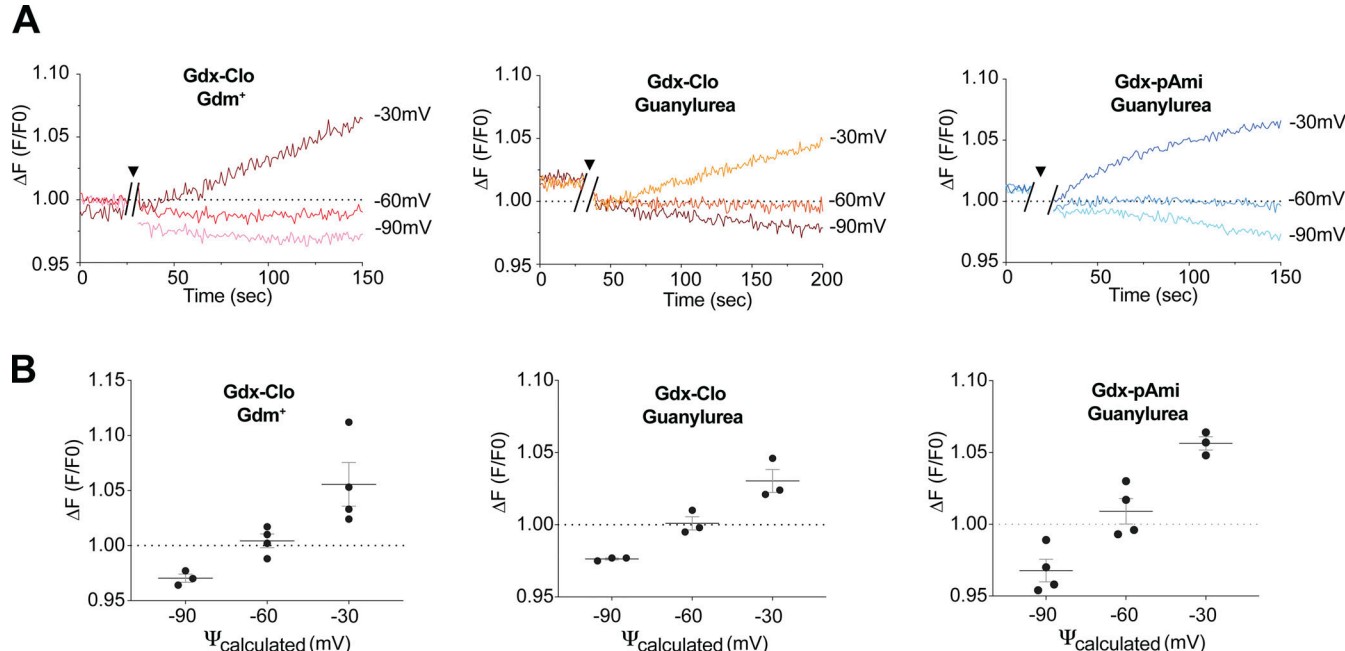

Figure 5. **Proton coupling stoichiometry for substrate transport by Gdx-Clo and Gdx-pAmi. (A)** Change in pyranine fluorescence over time for substrate transport at applied membrane potentials of −30, −60, and −90 mV. After ~20 s of baseline collection, the external substrate was added together with valinomycin to establish the 10-fold substrate gradient and membrane potential (indicated by break in trace and triangle). **(B)** Change in pyranine fluorescence as a function of membrane potential for replicate experiments. Measurements are a running 10 s average of the final 10 s of each trace (140–150 s for Gdx-Clo with Gdm+ and Gdx-pAmi with guanylurea, and 190–200 s for Gdx-pAmi with guanylurea). Error bars represent the SEM for three replicates (−90 and −30 mV) or four replicates (−60 mV). The dashed line represents the equilibrium condition where no proton transport occurs. $E_{rev}$ at −30 mV would correspond to a coupling ratio of 3 H+:1 solute, and $E_{rev}$ at −90 mV would correspond to a coupling ratio of 1.7 H+:1 solute (i.e., leaky transport).

evaluate differences in transport rate among different proteins (or different mutants of the same protein), we co-reconstituted each SMR$_{Gdx}$ homolog with an internal reference, the fluoride channel Fluc-Bpe (Stockbridge et al., 2013, 2015), so that both the test protein and the reference protein would be absorbed to the sensor in a prescribed molar ratio (Fig. 4 A). We selected Fluc-Bpe as an internal reference because of its extremely high selectivity for fluoride (McIlwain et al., 2021b) prevents cross-reactivity with other substrates or common buffer components. Moreover, its fast fluoride permeation rate and channel mechanism (McIlwain et al., 2021c) yield high sensitivity with small amounts of protein and low concentrations of fluoride. Control experiments with individually reconstituted Fluc-Bpe and SMR$_{Gdx}$ confirm that the SMR$_{Gdx}$ substrates guanidinium and guanylurea do not elicit a response from Fluc-Bpe, and that the SMR$_{Gdx}$ is similarly insensitive to fluoride perfusion (Fig. 4 B and Fig. S5). Between each substrate perfusion, we perfused with non-activating (substrate-free buffer) so that we could isolate the contribution of the Fluc or SMR$_{Gdx}$ to the current. By normalizing with respect to the peak fluoride current amplitudes, we obtain good sensor-to-sensor reproducibility (Fig. 4 C). At high protein concentrations or ion fluxes, the maximal currents can be limited by a number of factors such as internal volume, membrane potential, or membrane crowding. However, at the low protein concentrations used in these experiments (2.5 µg Gdx-Clo and 1 µg Fluc-Bpe per mg lipid), the normalized current amplitudes are reasonably linear with respect to the SMR$_{Gdx}$ concentration (Fig. 4 D), indicating that in the

concentration regime of these experiments, using Fluc-Bpe as a reference provides a linear readout of transport velocity (Fig. 4 D).

To assess the relative maximal transport velocities of the four SMR$_{Gdx}$ homologs, we evaluated the maximal (initial rate) capacitive currents upon perfusion with the substrate at a concentration fivefold higher than the $K_m$ values reported in Fig. 3. For each homolog, we independently assessed the reconstitution efficiency using quantitative Western blot analysis of liposomes (Fig. 4 E). Reconstitution efficiencies were similar for Gdx-Clo, Gdx-Eco, and Gdx-pPro (Table S2). For Gdx-pAmi, the reconstitution efficiency was ~10-fold lower. Co-reconstitution with Fluc-Bpe did not significantly change the reconstitution efficiency of any of the SMR$_{Gdx}$ homologs (assessed in triplicate, two-tailed t test: P = 0.22 for Gdx-Clo; P = 0.65 for Gdx-Eco; P = 0.93 for Gdx-pAmi; P = 0.74 for Gdx-pPro). Using peak fluoride current amplitudes as an internal reference, and adjusting for the measured reconstitution efficiency, these experiments show that the transport rates are comparable (within a factor of two) among the four SMR$_{Gdx}$. For Gdx-Clo, Gdx-Eco, and Gdx-pPro, the maximal velocity for Gdm+ is approximately twofold higher than for guanylurea, whereas, for Gdx-pAmi, the turnover rates of guanylurea and Gdm+ are comparable (Fig. 4 F).

The negative capacitive currents observed in the SSM electrophysiology experiments presented thus far are in accord with electrogenic transport of >1 H+ per substrate. Prior studies have shown that Gdx-Eco possesses a well-coupled 2 H+: 1 Gdm+ stoichiometry (Kermani et al., 2018; Thomas et al., 2021).

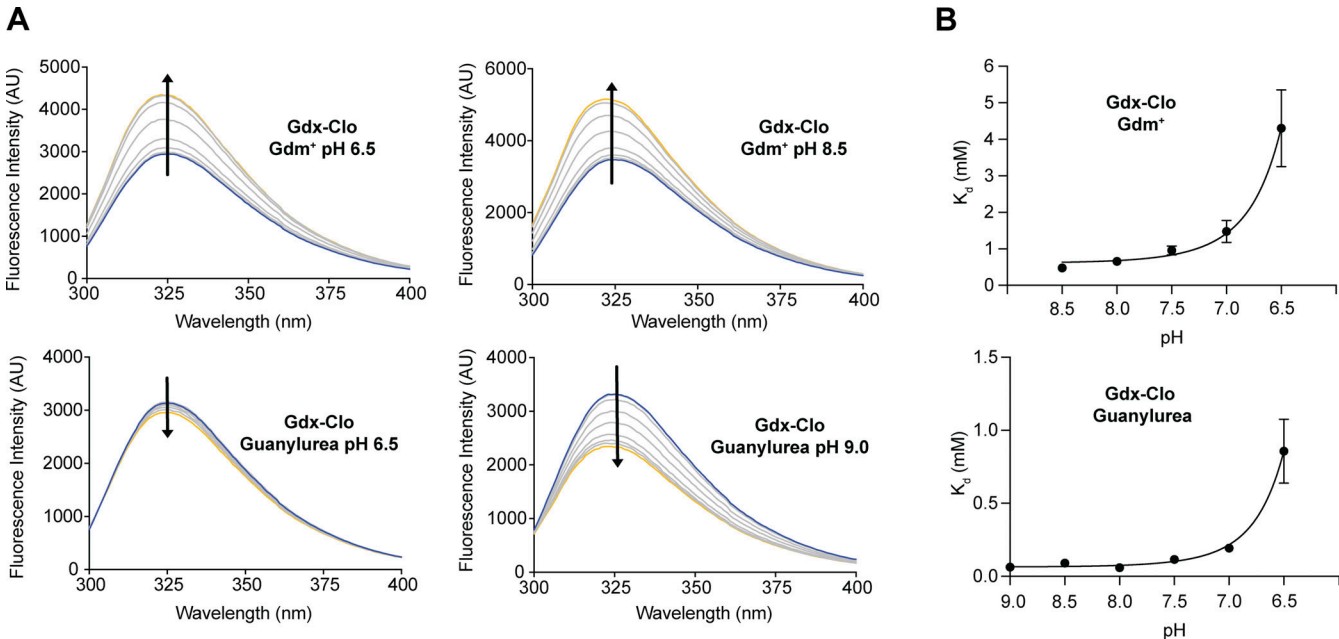

**Figure 6. pH dependence of equilibrium substrate binding for Gdx-Clo. (A)** Tryptophan fluorescence spectra were measured at increasing concentrations of Gdm$^+$ (top panels) or guanylurea (lower panels) at representative low and high pH values. Arrows denote the direction of change in fluorescence intensity with increasing substrate concentration. **(B)** The plot of apparent $K_d$ values measured for Gdm$^+$ (top) or guanylurea (bottom) as a function of pH. Apparent $K_d$ values were determined by fitting tryptophan fluorescence titration isotherms. Fluorescence spectra and fits for all pH values are shown in Fig. S5. The solid lines represent fits to Eq. 5, with a $K_d$ value of 600 µM and a $pK_a$ of 6.7 for the Gdm$^+$ titrations, and a $K_d$ value of 70 µM and a $pK_a$ of 6.9 for the guanylurea titrations. Error bars represent the SEM of values from three to four independent titrations from two independent protein preps.

However, for SMR$_{Qac}$ EmrE, it has been reported that the transport stoichiometry differs among some transported substrates (Robinson et al., 2017). We therefore employed a proteoliposome assay to experimentally assess the coupling stoichiometry of Gdx-Clo and plasmid-associated Gdx-pAmi. In these experiments, a 10-fold Gdm$^+$ or guanylurea concentration gradient is applied, and the direction of substrate movement is monitored as a function of membrane potential (Fitzgerald et al., 2017; Kermani et al., 2018). When no voltage is applied, the substrate is transported down its chemical gradient, coupled to proton efflux. Application of increasingly negative membrane potentials thermodynamically pushes back against the 10-fold substrate gradient; the electrochemical equilibrium point at which no substrate movement occurs is the reversal potential,

Table 2. **$K_d$ values for substrate binding to Gdx-Clo as a function of pH**

| pH | Gdm$^+$ (mM) ± SEM | Guanylurea (mM) ± SEM |
|---|---|---|
| 6.5 | 4.3 ± 1.0 | 0.86 ± 0.22 |
| 7.0 | 1.5 ± 0.3 | 0.19 ± 0.02 |
| 7.5 | 0.96 ± 0.12 | 0.12 ± 0.02 |
| 8.0 | 0.66 ± 0.08 | 0.059 ± 0.001 |
| 8.5 | 0.48 ± 0.10 | 0.091 ± 0.016 |
| 9.0 | Not determined | 0.063 ± 0.005 |

Values represent mean and SEM of three–four independent titrations from two independent protein preps.

from which the transport stoichiometry can be calculated using Eq. 2.

In our setup, the membrane potential is established using a potassium gradient and the potassium ionophore valinomycin, and substrate-coupled proton movement is monitored using pyranine, a pH-sensitive fluorescent dye, encapsulated inside the liposomes. With a 10-fold higher external solute, the electrochemical equilibrium is expected to occur at –60 mV for coupled 2 H$^+$: 1 solute transport. (Note that this value has the inverse sign—and is thus far from—the Nernstian reversal potential for solute of +60 mV that would be expected for uncoupled solute flux). For a coupling ratio of 3 H$^+$:1 solute, $E_{rev}$ would be equal to –30 mV, and for a leaky transporter with a reduced coupling ratio of 1.7 H$^+$:1 solute, $E_{rev}$ would be –90 mV. We examined proton flux at all three potentials for Gdx-Clo (Gdm$^+$ and guanylurea) and Gdx-pAmi (guanylurea only). For all three transporter and solute pairs examined, proton influx (decreased fluorescence) occurs at –90 mV and proton efflux (increased fluorescence) occurs at –30 mV. In contrast, at –60 mV, the fluorescence remains steady over the timecourse of the experiment, in agreement with a 2 H$^+$: 1 solute coupling stoichiometries (Fig. 5, A and B).

### Gdm$^+$ and guanylurea binding in Gdx-Clo
To further characterize the pH dependence and thermodynamic properties of Gdm$^+$ and guanylurea binding by SMR$_{Gdx}$, we selected the homolog with the best biochemical stability, Gdx-Clo. Although we initially sought to examine substrate binding by Gdx-pAmi as well, the protein requires high salt concentrations

Figure 7. **Isothermal titration calorimetry of Gdm+ and guanylurea binding to Gdx-Clo.** Top panels: Thermograms for Gdm+ titrations (left) and guanylurea titrations (right). Lower panels: Datapoints show heat absorbed as a function of substrate concentration, fit to equilibrium binding isotherms (solid lines). Traces are representative of three replicate experiments. Equilibrium binding parameters (mean and SEM) are shown in Table 3.

for purification and, in detergent, was prone to aggregate over long titrations or at more physiological salt concentrations.

We first exploited intrinsic changes in tryptophan fluorescence to monitor substrate binding at pH values between pH 6 and pH 9 (Fig. 6 A). Gdm+ titration induces an increase in tryptophan fluorescence that can be fit with a single site binding isotherm described by Eq. 3 (Fig. S6); separate control experiments showed that the binding reaction achieved equilibrium prior to measurement. As expected for a model where protons and Gdm+ compete for binding to the central glutamates, the apparent binding affinity increases with pH as the central glutamates become increasingly deprotonated (Table 2 and Fig. 6

Table 3. **Equilibrium binding parameters derived from isothermal titration calorimetry (pH 8.5)**

|  | Gdm+ | Guanylurea |
|---|---|---|
| $K_d$ (µM) | 410 ± 40 | 140 ± 30 |
| ΔG (kcal/mol) | −4.6 ± 0.1 | −5.4 ± 0.06 |
| ΔH (kcal/mol) | −4.9 ± 1.0 | −7.2 ± 0.6 |
| TΔS (kcal/mol) | −0.3 ± 0.9 | −1.8 ± 0.6 |
| n | 0.56 ± 0.04 | 0.43 ± 0.03 |

Mean and SEM from three independent experiments.

B). Although careful NMR experiments with SMR homolog EmrE have shown that the $pK_a$ values of the two central glutamates differ (Li et al., 2021; Morrison et al., 2015), the current binding assays do not have the resolution to distinguish independent $K_a$ values and report on the averaged behavior of the binding site residues. Using the approximation that the glutamates have equal $K_a$ values, the relationship between apparent $K_d$ and pH can be fit using Eq. 5, yielding an average $pK_a$ for the glutamates of 6.7 and a $K_d$ for Gdm+ of 600 µM. This value is in the same approximate range as the $pK_a$ values of the central glutamates in other SMR homologs (Li et al., 2021; Morrison et al., 2015; Muth and Schuldiner, 2000).

Analogous binding experiments were also performed for guanylurea (Fig. 6, lower panels). In contrast to the tryptophan fluorescence trend observed for Gdm+ binding, titration with guanylurea quenched the tryptophan fluorescence signal. Binding data also suggested there was also a low affinity, non-specific component to substrate binding, which became more apparent at high guanylurea concentrations. Fitting the data to a binding model with a linear non-specific component (Eq. 4) yields apparent $K_d$ values of the same order as the $K_m$ value determined previously. A fit to Eq. 5 indicates that the $pK_a$ value of the glutamates is 6.9, in reassuring agreement with the $pK_a$ determined in the Gdm+ binding experiment, and yields a guanylurea $K_d$ of 70 µM. For both substrates, the $K_d$ values are similar to the transport $K_m$ values,

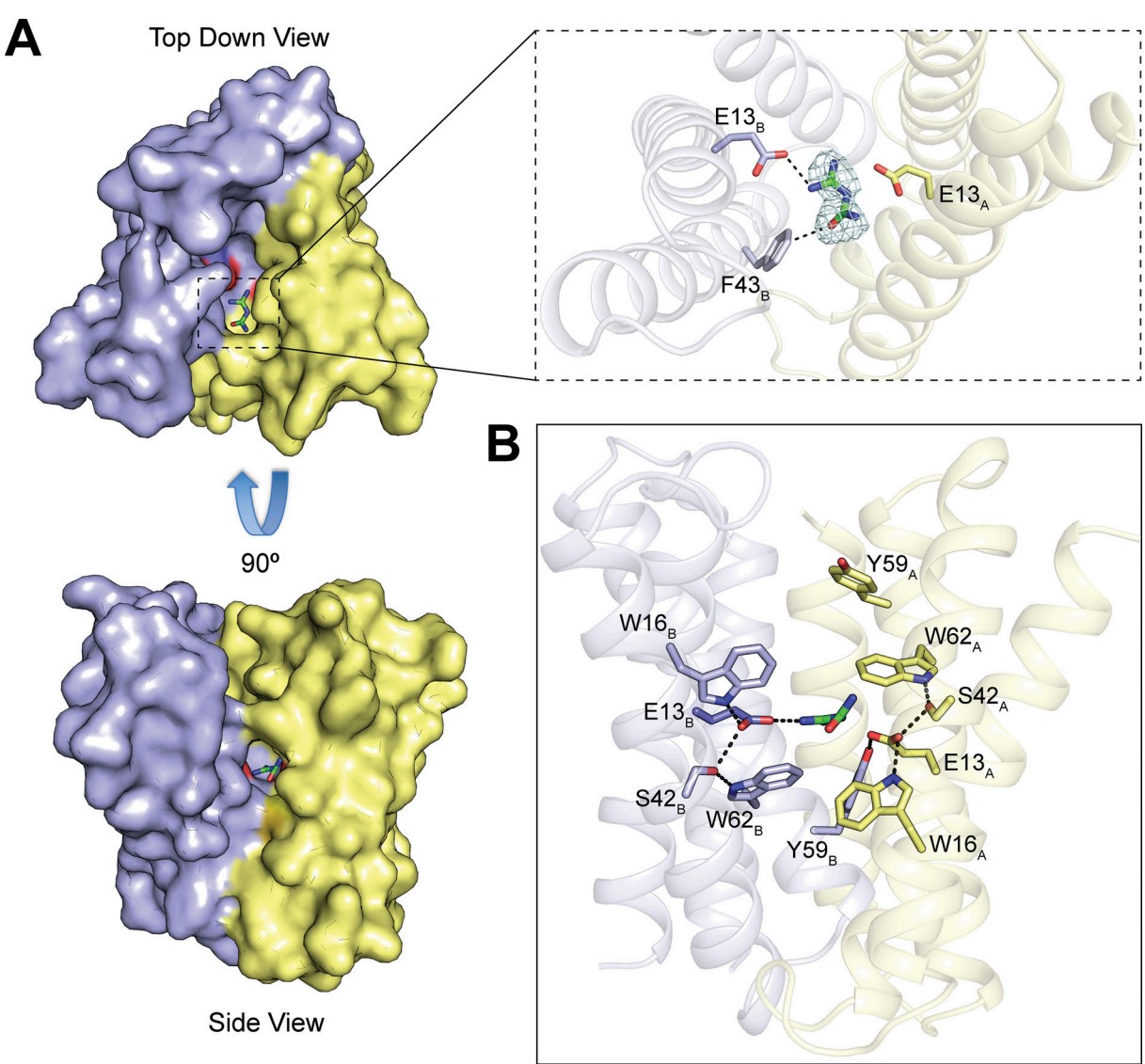

**Figure 8.** **Crystal structure of Gdx-Clo in complex with guanylurea. (A)** A and B subunits are shown in yellow and blue, respectively, with central glutamates shown as sticks and guanylurea as green sticks. The right upper panel shows the $F_o$-$F_c$ omit map for guanylurea contoured at 3.5σ. **(B)** Putative polar interactions among binding site residues are indicated with dashed lines.

suggesting that the kinetics of substrate binding are fast relative to the conformational change during substrate transport.

Because tryptophan fluorescence is an indirect measurement of binding (made additionally mysterious by the opposite effects of Gdm+ and guanylurea on the fluorescence intensity), we also sought to reproduce our binding measurements using ITC. At pH 8.5, where proton binding to the glutamates is minimized, we observed an exothermic binding reaction for both Gdm+ and guanylurea with the expected stoichiometry of ~1 substrate per protein dimer (Fig. 7 and Table 3). For both substrates, the $K_d$ value measured using ITC was in good agreement with the $K_d$ value obtained using tryptophan fluorescence, validating the tryptophan fluorescence approach to monitor substrate binding. The approximately threefold increase in affinity for guanylurea relative to Gdm+ was due to a more favorable enthalpy of the binding reaction. Thermodynamic parameters derived from the ITC data are reported in Table 3.

Finally, to determine whether guanylurea occupies the same binding pocket as guanidinium in Gdx-Clo, we solved a crystal structure of Gdx-Clo in the presence of 10 mM guanylurea (Fig. 8 and Table 4). Crystals were prepared as in previous studies (Kermani et al., 2020, 2022) and diffracted to 2.1 Å. Two transporters are present in the unit cell, and the maps showed clearly resolved guanylurea density nestled in the binding pocket of one of these transporters (Fig. 8 A). The guanidinium group is poised between the central glutamates, within hydrogen bonding distance, in the same binding mode as observed for phenylguanidinium (Kermani et al., 2020). The carbonyl of guanylurea faces the cleft between helices $2_A$ and $2_B$ (termed the hydrophobic portal [Kermani et al., 2020]), but is just small enough to fit in the binding pocket without requiring a rearrangement of the sidechains lining the portal, in contrast to the slightly larger phenylguanidinium (Kermani et al., 2020). The carbonyl of the guanylurea is twisted slightly out of plane with

**Table 4. Data collection and refinement statistics for the crystal structure of Gdx-Clo in complex with guanylurea**

| Data collection | |
|---|---|
| Space group | P1 |
| Cell dimensions $a$, $b$, $c$, (Å) | 49.91, 74.37, 107.21 |
| α, β, γ (°) | 86.76, 90.01, 70.26 |
| Resolution (Å) | 107–2.185 |
| Ellipsoidal resolution limit (best/worst) | 2.185/3.665 |
| % Spherical data completeness | 39.0 (5.6) |
| % Ellipsoidal data completeness | 81.1 (69.8) |
| $R_{merge}$ | 0.600 (0.099) |
| $R_{meas}$ | 0.698 (0.122) |
| Mn $I/\sigma I$ | 18.1 (2.8) |
| Multiplicity | 3.3 (3.8) |
| **Refinement** | |
| Resolution (Å) | 35.66–2.18 |
| No. reflections | 27,820 |
| $R_{work}/R_{free}$ | 28.0/30.9 |
| Ramachandran favored | 94.7 |
| Ramachandran outliers | 1.8 |
| Clashscore | 11.2 |
| **R.m.s. deviations** | |
| Bond lengths (A) | 0.002 |
| Bond angles (°) | 0.480 |
| Coordinates in PDB | 8TGY |

respect to the guanidinyl group and is positioned ∼3 Å from the electropositive ring edge of portal sidechain F43. There are no other residues within the coordination distance of guanylurea, recapitulating the undercoordination of the native substrate Gdm$^+$. Other key binding pocket residues (W16, S42, Y59, and W62) contribute to an H-bond network that stabilizes the central E13 residues in the same orientation as seen in other structures (Fig. 8 B; Kermani et al., 2020, 2022).

## Discussion

Microbes are constantly evolving to contend with new environmental pressures, including the recent introduction of anthropogenic chemicals. Major routes for the acquisition of new traits by a microbial population include the gain of new genes via HGT transfer events and the co-option of native proteins' cryptic functions (functions not under natural selection) to fulfill novel functional roles. Here, we examine a family of transporters, the SMRs, that are associated with both evolutionary processes. In particular, we focus on the SMR$_{Gdx}$, which undergoes frequent HGT, despite playing little role in bacterial resistance to classical antimicrobials or antiseptics (Kermani et al., 2020; Slipski et al., 2020). Based on genetic evidence (Li et al., 2023; Martinez-Vaz et al., 2022), we hypothesized a role for the SMR$_{Gdx}$ in the transport of metformin metabolites, which

structurally resemble the native substrate Gdm$^+$, and have accumulated to high levels in waste and surface waters. Our previous work provided preliminary support for this possibility (Martinez-Vaz et al., 2022).

In this study, we investigate whether the export of guanylurea or other metformin metabolites is a general property of SMR$_{Gdx}$, and we functionally characterize this activity across multiple plasmid-associated and genomic transporters. We show robust transport of guanylurea, with the same transport stoichiometry, and transport kinetics in the same order as that of the physiological substrate Gdm$^+$. Structures of the guanylurea-bound transporter Gdx-Clo show how guanylurea binding exploits the protein's under coordination of the native substrate, Gdm$^+$ (Kermani et al., 2020), fulfilling all of the hydrogen bonds seen for the native substrate without interference from the substrate's urea group.

It was surprising on its face that the homolog with the most explicit connection to metformin degradation, Gdx-pAmi, had the lowest affinity for guanylurea (5 mM). But for bacteria actively metabolizing metformin as a nitrogen source, very high concentrations of guanylurea are likely to be produced. A prior study measured metformin degradation by an *Aminobacter* culture at a rate of ∼0.7 mM/h (Li et al., 2023). Considering the culture density and approximating a ∼femtoliter volume for each cell, each bacterium will produce nearly 16 mM internal guanylurea per minute. This back-of-the-envelope calculation illustrates the need for an efflux pathway and also suggests that bacteria that occupy this niche might be adapted to handle high steady-state guanylurea concentrations. It is a truism that an enzyme only needs to be good enough, and apparently, high substrate affinity is not essential for Gdx-pAmi to contribute a selective advantage in the context of metformin degradation.

In summary, this work has functionally characterized an emerging physiological role of the SMR$_{Gdx}$ transporters for the export of metformin metabolites. Such a function rationalizes their genetic occurrence with wastewater-associated plasmids and may also have implications for species distribution or horizontal gene transfer in the gut microbiome of patients treated with metformin. Moreover, understanding how bacteria co-opt native physiologies to contend with novel xenobiotics yields insights into microbial adaptation to an increasingly human-impacted biosphere. Our current study highlights a role for active transport in the full microbial degradation pathway for a chemical pollutant and may inform effective multispecies bioremediation strategies for metformin and other pharmaceuticals in the environment.

### Data availability

Atomic coordinates for Gdx-Clo bound to guanylurea have been deposited in the Protein Data Bank under accession numbers 8TGY. Source data for figures is available from the corresponding author upon request.

### Acknowledgments

Joseph A. Mindell served as editor.

The authors thank Chia-Yu Kang for assisting with MPER-tagged Fluc-Bpe.

This work was supported by National Institutes of Health grant R35-GM128768 to R.B. Stockbridge and resources of the Advanced Photon Source, a U.S. Department of Energy (DOE) Office of Science User Facility operated for the DOE Office of Science by Argonne National Laboratory under contract no. DE-AC02-06CH11357. Use of the LS-CAT Sector 21 was supported by the Michigan Economic Development Corporation and the Michigan Technology Tri-Corridor (grant 085P1000817).

Author contributions: R.M. Lucero: Conceptualization, investigation, writing—original draft, visualization; K. Demirer: Conceptualization, investigation, validation, writing—original draft, visualization; T.J. Yeh: Conceptualization, investigation, methodology, writing—review and editing, visualization; R.B. Stockbridge: Conceptualization, investigation, writing—original draft, writing—review and editing, funding acquisition, project administration. R.M. Lucero and K. Demirer characterized all SMR$_{Gdx}$ and SMR$_{Qac}$ homologs. T.J. Yeh developed and analyzed the Gdx-Clo A70C/S104C crosslink and developed the sensor normalization approach using Fluc-Bpe.

Disclosures: The authors declare no competing interests exist.

Submitted: 14 August 2023

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

## Supplemental material

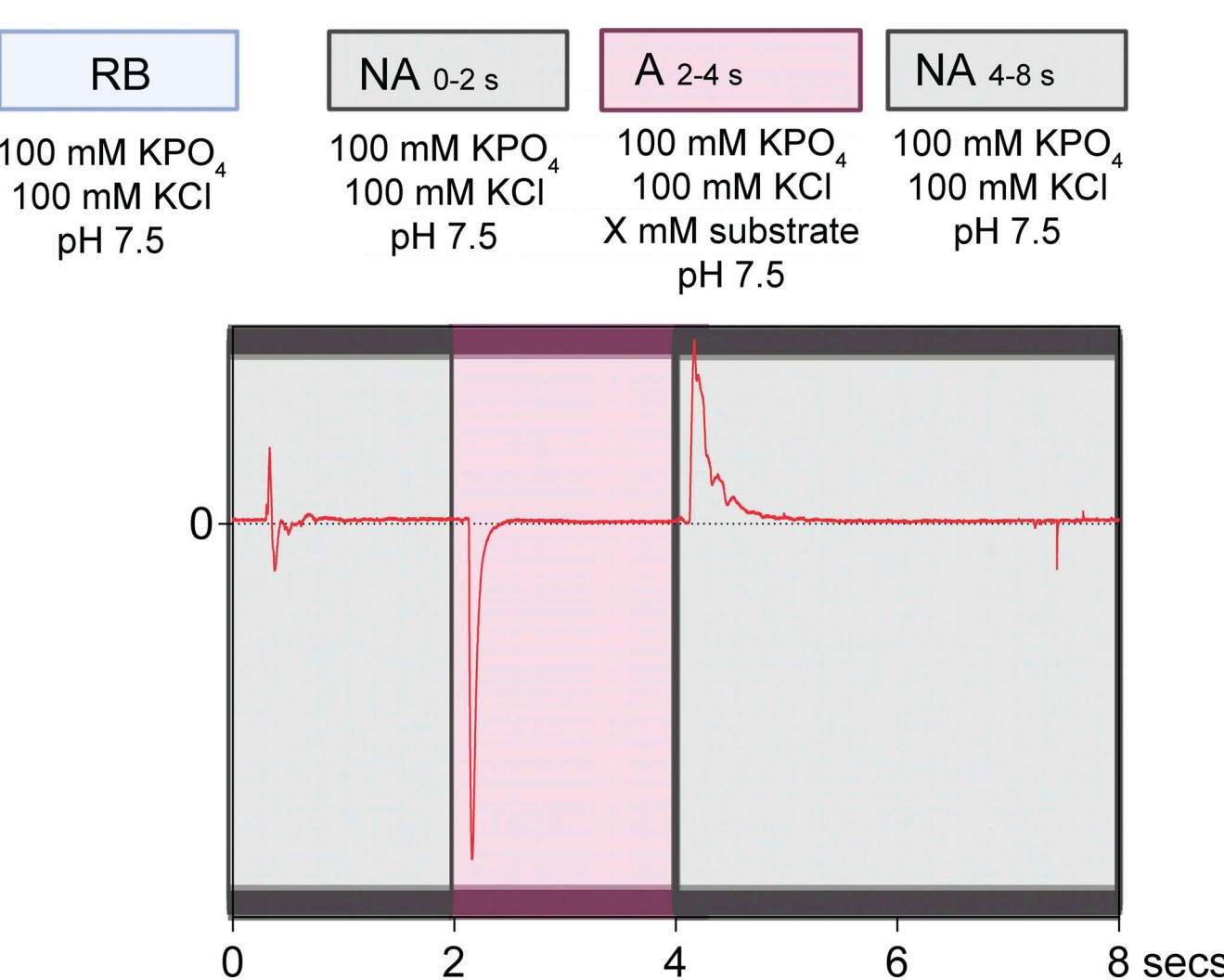

Figure S1. **Solution exchange protocol and example trace for SSM electrophysiology experiments described in this manuscript.** An example of a full SSM electrophysiology protocol showing perfusion with non-activating buffers (gray) and substrate (pink). RB stands for "reconstitution buffer," which is the non-activating buffer used for these experiments.

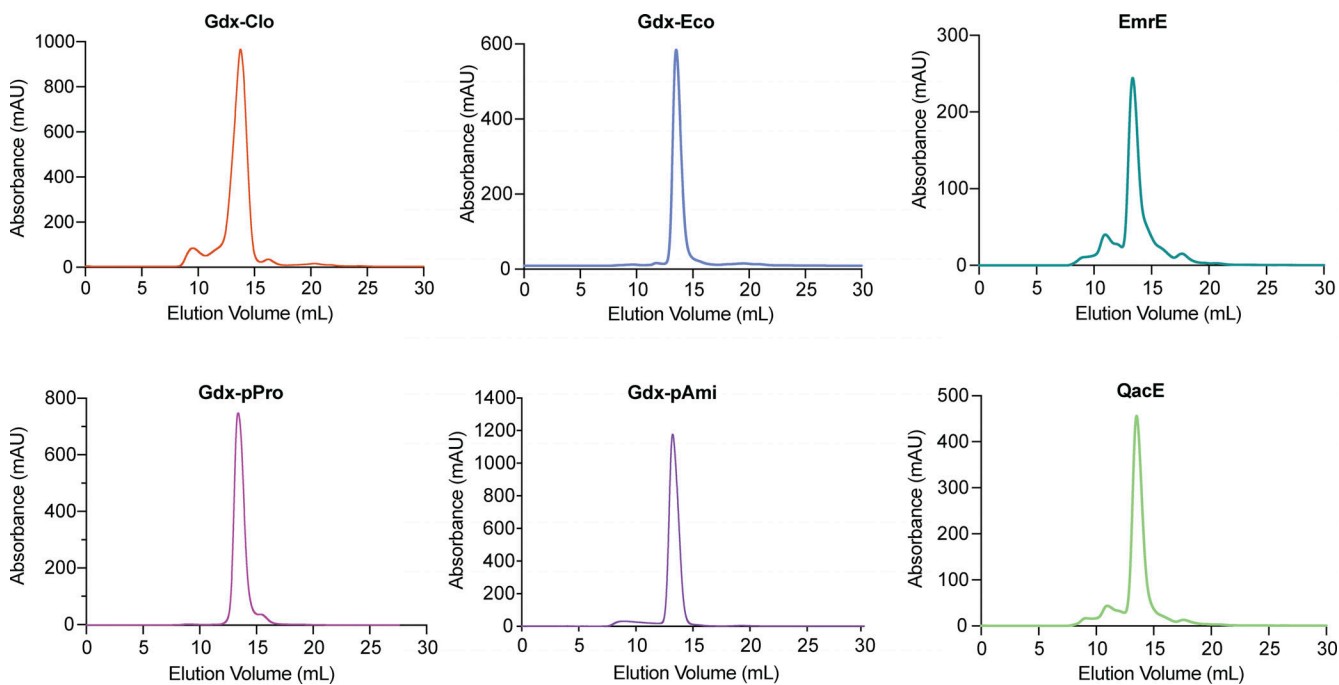

Figure S2. **Size exclusion chromatograms for six proteins in this study.** The major peak was collected for biochemical analysis.

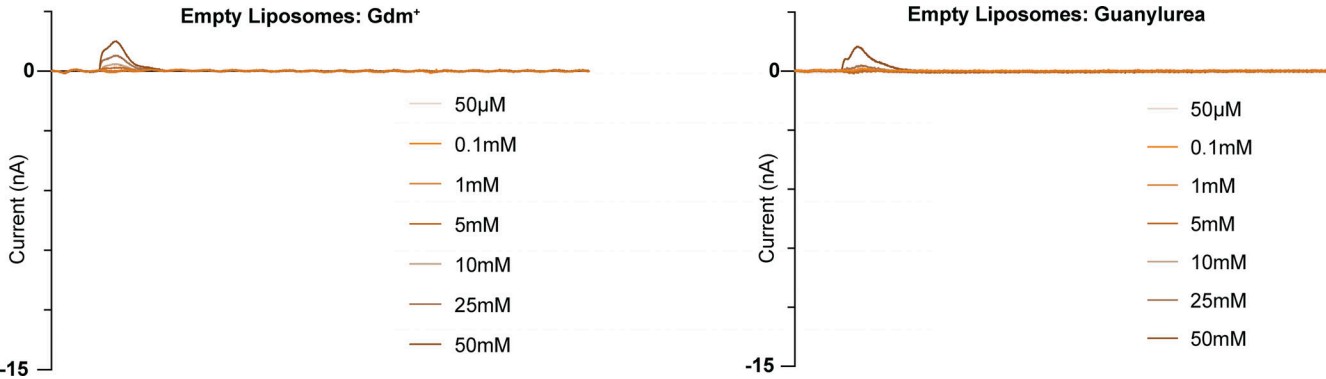

Figure S3. **Representative current traces for substrates and transporter data are summarized in** Fig. 1**, and no-protein controls.** Only the substrate perfusion step is shown. Traces for a substrate series are from the same sensor. Box height is equal to current values shown at right. **(A–C)** The panels in A show currents for SMRGdx homologs, the panels in B show currents for SMRQac homologs, and panels in C show the protein-free liposome control experiments.

Figure S4. **Representative current traces for Gdm⁺ and guanylurea titrations of protein-free liposomes.** Traces from a single sensor. Only the substrate perfusion step is shown.

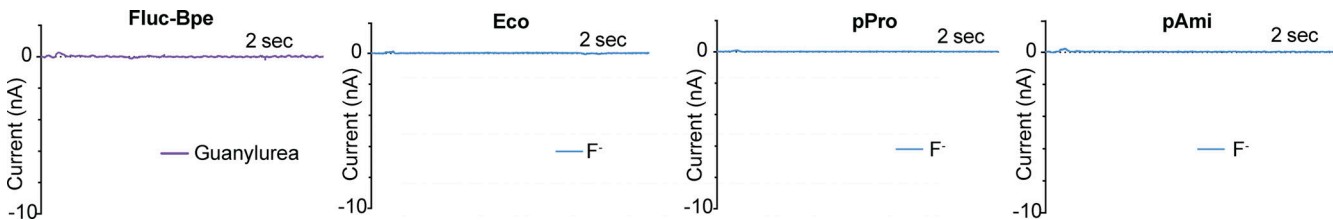

Figure S5. **Representative current traces for guanylurea perfusion of Fluc-Bpe and fluoride perfusions of Gdx-Eco, Gdx-pPro, and Gdx-pAmi.** Only the substrate perfusion step is shown.

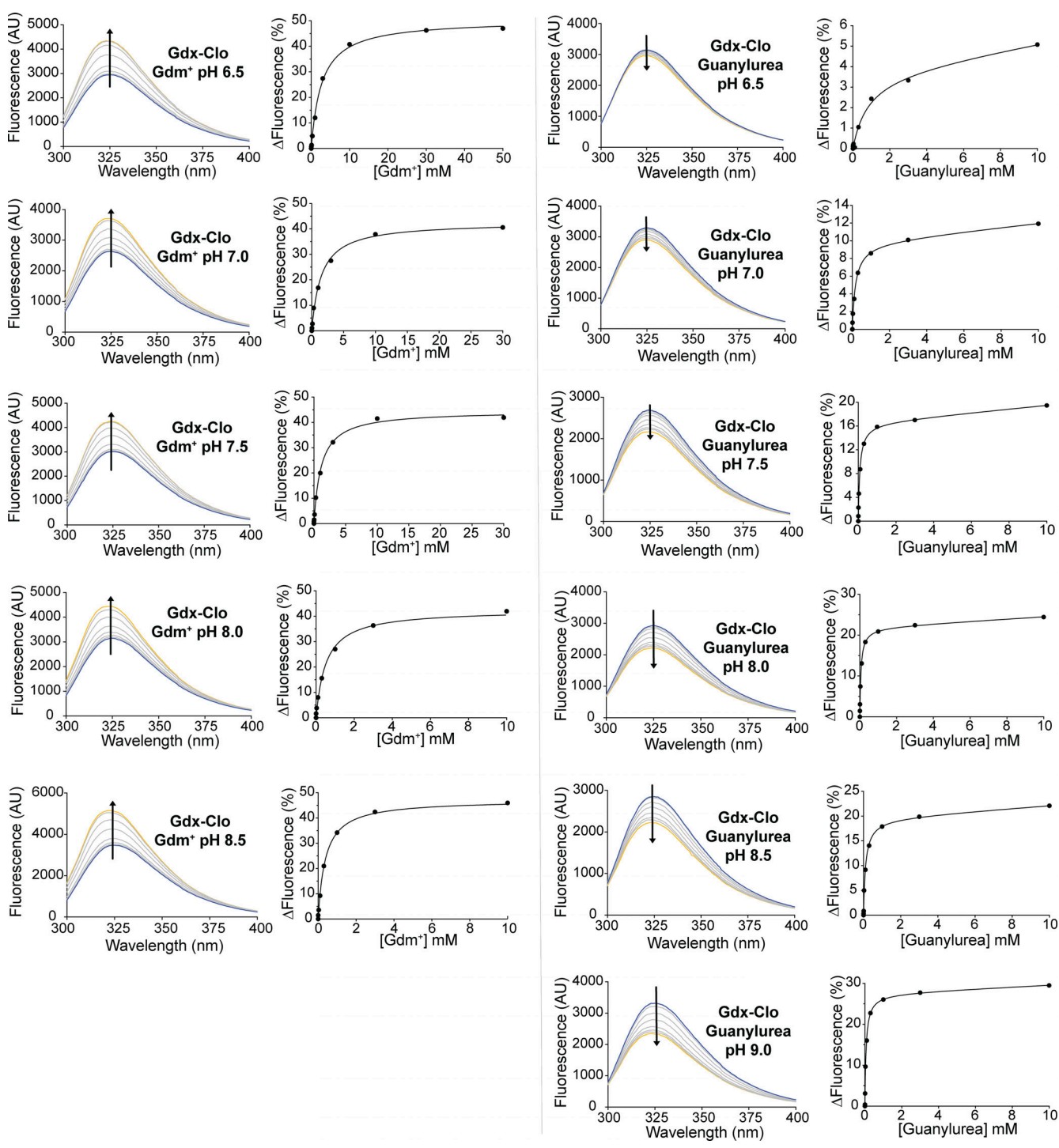

**Figure S6. Tryptophan fluorescence spectra and fits to binding isotherms for all data reported in** Fig. 6 **and** Table 2**.** For the spectra of each representative titration series, the arrow indicates whether fluorescence increases or decreases with increasing substrate. Representative plots of substrate concentration versus change in fluorescence (from a single titration) are also shown and are fit to a single-site binding isotherm (for Gdm$^+$ titrations) or a single-site binding isotherm with a linear correction for non-specific binding (for guanylurea titrations) as described in the Materials and methods.

Provided online are Table S1 and Table S2. Table S1 shows coding sequences for transporters examined in this study. Table S2 shows reconstitution efficiencies of SMR$_{Gdx}$ homologs assessed by quantitative Western blot.

