## [Peer Review File · The Journal of General Physiology]

Transport of metformin metabolites by guanidinium exporters of the Small Multidrug Resistance family

Rachael Lucero, Kemal Demirer, Trevor Yeh, and Randy Stockbridge

Corresponding Author(s): Randy Stockbridge, University of Michigan-Ann Arbor

Review Timeline:

Submission Date:	August 14, 2023
Editorial Decision:	September 13, 2023
Revision Received:	December 1, 2023
Editorial Decision:	December 20, 2023
Revision Received:	December 22, 2023

Editor: Joseph Mindell

Transaction Report:

DOI: <https://doi.org/10.1085/jgp.202313464>

September 13, 2023

Prof. Randy B. Stockbridge
University of Michigan-Ann Arbor
MCDB and Biophysics
1105 N. University Ave.
Biological Sciences Building Room 3242
Ann Arbor, MI 48109

Re: 202313464

Dear Randy,

Thank you for submitting your manuscript, entitled "Transport of metformin metabolites by guanidinium exporters of the Small Multidrug Resistance family" to JGP. Your manuscript has now been seen by 3 reviewers, whose comments are appended below. You will see that the reviewers were very enthusiastic about the study and its potential impact though they did raise a series of concerns that need to be addressed. In particular, both Reviewer #1 and Reviewer #3 raise important issues about whether the Fluc normalization experiments really can be effective at their aim. Notably also, Reviewer #3's comment #1 about adsorption vs fusion not only applies to the methods but to multiple instances in the results that also need to be corrected.

In addition to the reviewers' concerns, one issue came up in my own reading of the manuscript regarding the stoichiometry measurements. First, I think there is an error in equation 2, and that the numerator should be m , not n (i.e. $m/(m-n)$). Second, and more important, an odd situation arises because you have no pH gradient, and because of the 2:1 stoichiometry of these proteins. In this case, $m/(m-n)$ is -1 , and so E_{rev} is equal to the Nernst potential for substrate (at least in absolute value), which would also be true if you had an uncoupled channel. In your case, though, since you're actually measuring the reversal of substrate-induced proton flux I think your data indeed support your conclusion-but I think its worth a few words of discussion to clarify for a reader who might be confused seeing Nernstian reversal for a transporter. In my lab we always also introduce a pH (or other driving ion gradient) in such cases to ensure that the predicted E_{rev} is far from Nernst... I think it would also be helpful to indicate on the reversal potential graph the expected values for alternate stoichiometries like 3:1 or 1:2...

Also, though your data are nicely presented (love that you're showing all the points for repeated measurements) and the effects you are observing seem large, we still expect an appropriate statistical measure to support your interpretation of differences. I expect that your P-values will be quite low, but you do need to calculate and display them.

We hope that you will be able to submit a revised manuscript that addresses these points, which we believe will pose no problems, and which may be re-reviewed. Based on the scope of the requested changes, we typically anticipate that the revision process will take no longer than 2 months, however, we understand you may need additional time to work on your resubmission to JGP. We therefore ask that you simply keep us informed as to a realistic submission timeline that is appropriate for your particular circumstances. In addition, please do not hesitate to contact me (via the editorial office) if you feel that a discussion of the reviewers' and editors' comments would be helpful.

Please submit your revised manuscript via the link below, along with a point-by-point letter that details your response to the reviewers' and editors' comments, as well as a copy of the text with alterations highlighted (boldfaced or underlined). If the article is eventually accepted, it would include a 'revised date' as well as submitted and accepted dates. If we do not receive the revised manuscript within one year, we will regard the article as having been withdrawn. We would be willing to receive a revision of the manuscript at a later time, but the manuscript will then be treated as a new submission, with a new manuscript number.

Please pay particular attention to recent changes to our instructions to authors in the following sections: Data presentation, Blinding and randomization and Statistical analysis, under Materials and Methods, as shown here: <https://rupress.org/jgp/pages/submission-guidelines#prepare>. Re-review will be contingent on inclusion of the required information (including for data added during revision) and demonstration of the experimental reproducibility of the results. Also, To improve the reproducibility of published content, we have partnered with SciScore. Authors are prompted in eJP to copy and paste the Materials and Methods section of their manuscript for a SciScore assessment when submitting their revised manuscript. Authors are encouraged (not required) to further revise their Materials and Methods if the SciScore is below 4. More information can be found here: <https://rupress.org/jgp/pages/submission-guidelines#sciscore>.

Please note, JGP now requires authors to submit Source Data used to generate figures containing gels and Western blots with all revised manuscripts (when applicable). This Source Data consists of fully uncropped and unprocessed images for each gel/blot displayed in the main and supplemental figures. If your paper includes cropped gel and/or blot images, please be sure to

provide one Source Data file for each figure that contains gels and/or blots along with your revised manuscript files. File names for Source Data figures should be alphanumeric without any spaces or special characters (i.e., SourceDataF#, where F# refers to the associated main figure number or SourceDataFS# for those associated with Supplementary figures). The lanes of the gels/blots should be labeled as they are in the associated figure, the place where cropping was applied should be marked (with a box), and molecular weight/size standards should be labeled wherever possible.

Source Data files will be made available to reviewers during evaluation of revised manuscripts and, if your paper is eventually published in JGP, the files will be directly linked to specific figures in the published article.

Source Data Figures should be provided as individual PDF files (one file per figure). Authors should endeavor to retain a minimum resolution of 300 dpi or pixels per inch. Please review our instructions for export from Photoshop, Illustrator, and PowerPoint here: <https://rupress.org/jgp/pages/submission-guidelines#revised>

Whilst you are revising your manuscript, we ask that you consider whether you have any artwork that might be suitable for the cover of JGP. Microscopy images are particularly good for cover artwork, but other types of image can be very effective, so we encourage you to be creative. Please don't restrict yourself to images from the paper; an image that is relevant to the work described would be just as suitable. Images should be a minimum resolution of 300 dpi. To see recent examples, visit the following page and click on 'Show covers? Yes': <https://jgp.rupress.org/content/by/year>

Thank you for submitting your interesting research to JGP.

Please submit your revised manuscript, and any associated files, via this link:

Link Not Available

Sincerely,
Joe

Joseph A. Mindell, M.D., Ph.D.
On behalf of Journal of General Physiology

Journal of General Physiology's mission is to publish mechanistic and quantitative molecular and cellular physiology of the highest quality; to provide a best-in-class author experience; and to nurture future generations of independent researchers.

Reviewer #1 (Comments to the Authors):

Membrane transporters belonging to the Small Multidrug Resistance (SMR) family are found in bacteria and archaea. They play a crucial role in microbial resistance by exporting toxic compounds like antibiotics from the cell. Guanidinium exporters (Gdx) are part of the SMR family of proteins. While SMRGdx transporters don't provide antibiotic resistance, a recent study suggests their involvement in the bacterial metabolism of the widely used diabetes drug metformin and its metabolite guanylurea. This creates new selective pressures for bacterial communities with unknown consequences.

In this study, Lucero and co-authors explore the impact of metformin metabolites on the function of four SMR-Gdx homologues. Using solid-supported membrane (SSM) electrophysiology, they discovered that, out of the seven studied substrates, guanylurea induces protein-mediated current amplitudes comparable with, or even greater than, those induced by guanidinium (Gdm⁺). They find that for all homologues the Michaelis-Menten Constants (K_m) varied by factor of ~50, while for the same homologue the K_m of guanylurea was always ~2-fold lower than that of Gdm⁺. By co-reconstitution of the fluoride channel Fluc into liposomes as an internal control, they attempt to compare the transport rates of the four different homologues and two substrates with SSM. Furthermore, the authors use a liposome-based pyranine assay to reveal a 1:2 substrate to proton stoichiometries. Here, the authors follow a substrate proton exchange at given membrane potentials using the pyranine-filled proteoliposomes excite at 455 nm and measure the emission at 515 nm. Furthermore, the authors focused on one of four homologues to study and compare the binding and pH dependence of Gdm⁺ and guanylurea. For this purpose, they use both isothermal titration calorimetry (ITC) and tryptophan fluorescence and find comparable K_d values, similar to the K_m values obtained through SSM. They conclude their investigation by determining the crystal structure of the chosen homologue in the presence of guanylurea, which take the same mode as observed before.

I appreciate the clear language in the manuscript and the combination of different functional methods the authors used to learn more about the SMRGdx function. Even though they carefully included controls in many of their experiment, a few study-related issues deserve notice.

Major:

1. The authors attempt to use SSM electrophysiology to quantitatively compare transport rates of the four different SMRGdx homologues and two substrates. A quantitative comparison is usually impossible because of the fluctuation in current measurements between liposome prep, sensors, or even day-to-day experiments. The authors aim to circumvent this problem

by co-reconstituting fluoride channel Fluc and using its currents as an internal standard. While I find this approach intriguing, I see two issues in the results shown in Figure 4.

The first issue is the absence of a non-activating solution in their solution exchange protocol (Figure 4A). This absence creates a problem because switching from fluoride directly to Gdm⁺ activates two processes: the efflux of the taken-up fluoride through Fluc and the exchange of Gdm⁺ and protons through SMRGdx. This might compromise the accuracy of transport rate comparisons. If non-activating solutions were indeed used, it is important to specify this in the methods section and perhaps the cartoon depiction in Figure 4A.

Furthermore, if SSM is used to compare the differences in transport rates among the four homologues, it is essential to show consistent reconstitution efficiencies between all co-reconstituted proteins. If the authors did not intend for the homologues to be compared but just the two substrates, I suggest avoiding normalizing to a single homologue reference (Gdx-Clo). Otherwise, I propose including quantitative Western Blot analysis to assess the reconstitution efficiency of Fluc and all co-reconstituted proteins.

2. The authors use a liposome-based pyranine assay to determine a substrate-to-proton stoichiometry of 1:2 for guanylylurea and Gdm⁺. However, the results presented in Figure 5A need clarification.

How was the baseline fluorescence for the experiments defined by the authors? There appears to be an offset in fluorescence at baseline before the addition of guanylylurea and valinomycin for the Gdx-Clo and Gdx-pAmi liposome samples. It is also unclear why the authors did not use the fluorescence ratio of pyranine by exciting at two instead of only one wavelength.

Additionally, it would enhance the study to include control experiments with no substrate but valinomycin and vice versa.

Also, the authors should indicate the exact time point from the kinetic experiment shown in Figure 5A that was used for the graphs in Figure 5B. Including this information in Figure 5 would improve the clarity.

Minor:

3. If the authors could perform the ITC experiments in triplicates, this would considerably increase the robustness of their results. This becomes more important considering that, despite careful controls, fluctuations in the buffer composition could be the cause of the endothermic contribution reported during guanylylurea titration.

4. Furthermore, the authors should expand the ITC method section to include critical details such as titration volume, interval time, and differential power settings.

5. Please provide more details for the SSM method section, such as the description of solution exchange protocols, primarily if they are not described in the figure legends.

6. Tables 1 and 2 should contain legends to display how many technical or replicates they encompass.

7. Figure 8 would benefit from also including the label "Side View" to more easily identify the different perspectives.

8. Provide a proper figure legend for the Supplementary Figure to offer the reader a clear understanding of the content in the main text.

Reviewer #2 (Comments to the Authors):

The manuscript from Stockbridge's lab describes the results of an elegant study designed to identify the role of the Small Multidrug Resistance (SMRs) family of proton-coupled antiporters on the transport of Metformin or other byproducts of microbial metformin metabolism. This is a relevant question since Metformin is a frequently prescribed drug to treat type II diabetes; it is dosed in gram quantities and is excreted in an unaltered form. It has, therefore, an impact on the environment and the human microbiome.

The authors identified SMRGdx (one of the subgroups in the SMR family) homologs that transport Guanylylurea, a byproduct of microbial metformin metabolism. They showed that efficient guanylylurea transport is a general property of the SMRGdx subtype but not of SMRQac and that SMRGdx also transports other metformin degradation products.

They then characterized the transport kinetics and proton-coupling stoichiometry of a representative plasmid-borne and genomic SMRGdx. They determined the structure of Gdx-Clo, the well-characterized genomic protein from Clostridia, with guanylylurea bound.

The study is relevant, well-planned, and carried out. The paper would significantly benefit from editorial work to produce a leaner version. Some paragraphs are repetitive and describe in unnecessary detail the rationale for experimental design (e.g., p.9: "Our experiments thus far do not..."; p. 11: "Although we initially sought..."; p.12: "Because tryptophan fluorescence; etc.). Is there a reason to go into so much detail describing methods previously published or writing the Nernst equation?

Specific points:

p.11: The authors state that separate control experiments showed that binding kinetics are fast and that the binding reaction achieved equilibrium before measurement. What is fast? Usually, binding per se is a very fast reaction, not just fast. If this is not the case for the proteins studied, it may indicate some slow conformational changes required before binding. Is this the case?

p. 10: Isn't stoichiometry for EmrE 2:1 also for most classical substrates? The way it is written sounds like varying stoichiometries is this transporter's signature.

How does the Glutamate pK compare with that of EmrE? I guess there were studies other than the NMR ones.

Reviewer #3 (Comments to the Authors):

In the era of structural characterization, detailed functional characterization often comes too short. But it is very much required to actually confirm the structure-based functional models. In this manuscript the authors contribute important kinetic data for proteins of the SMR family, even allowing a kinetic comparison of different homologues, that may look pretty much the same in terms of structure.

Besides investigations on transporter kinetics in general, the authors casted light on the physiological relevance of these transporters based on their findings and focused on the very relevant substrate metformin, the most frequently prescribed drug worldwide, which is also an issue due to wastewater contamination. SMR proteins seem to play a key role in metformin degradation, by exporting intermediates.

Another plus is the use of many different biochemical approaches, which makes the presented data very convincing.

Please check the following comments containing both, corrections and suggestions to further improve the manuscript.

1. In the methods section about SSME, "prior to fusion with DPhPC monolayer" => proteoliposomes do not fuse with the SSM, but they physically adsorb or attach.
2. In the same section, what is meant by positive reference samples? Is a reference substrate used to activate the transporter within the same sensor after five measurement of different substrates to check stability? Maybe clarify a little here and if the "reference sample" means substrate, indicate the reference substrate as well.
3. Methods section, Pyranin Assay: "1mM Pyranine" - space is missing between 1 and mM; "~30 s. to" - replace dot by comma?
4. Methods section, Tryptophan Fluorescence: "1µM purified...", "5mM DM" - space missing;
5. Method section, structure: "Gdx-Clo (10mg/mL) and L10 monobody (10mg/mL," - space missing two times
6. Minor concern about data visualization in Figure 2: "Datapoints represent at least three independent sensor preparations from at least two independent biochemical purifications." Am I understanding correctly, that the shown datapoints, averages and SEMs consider data from the same sensor, different sensors and different biochemical purifications at the same time? I don't feel well with this kind of representation/analysis, since usually datapoints from the same sensor show much lower variations compared to analyzing different sensors. And different biochemical purifications may show even higher variations in terms of amplitude. Typically I would only use one datapoint per sensor (or the average across all datapoints recorded within the same sensor) and subsequently average across sensors. Data from different biochemical purifications may be highlighted by color, but since the plot shows normalized data, I do not expect large differences between purifications beyond sensor-sensor variations.
7. Page 9: "The current amplitudes reflect the initial rate of transport(Bazzone et al., 2017), and their concentration dependence follows Michaelis-Menten kinetics (Figure 3, Table 1)." - this is true, providing a certain line of evidence, i.e. absence of pre steady-state currents. Two more recent publications from 2023 focus on this topic, which would be better citations: 10.1016/j.bios.2021.113763, 10.3389/fphys.2023.1058583 (the second one actually highlights and compares analysis methods to distinguish transport from binding events recorded with SSME).
8. In the same context: Maybe also highlight and cite the evidence for SMR proteins to show only transport currents - I'm sure that was discussed in one of the authors earlier publications.
9. Later on page 9: "We verified that protein-free liposomes did not exhibit currents at all substrate concentrations tested (Supplementary Figure 3)." - there are small currents at the highest concentrations used. I would be more defensive in this sentence, like "only the highest substrate concentrations show small currents, in the range of 10. Last paragraph, page 9: "As with other measurements that rely on liposome fusion to a lipid bilayer(Stockbridge and Tsai, 2015),..." - SSME does not rely on the fusion, but adsorption as pointed out later in the same sentence. However, later "fusion efficiency" is mentioned. I would consistently say "adsorption", if possible. Fusion is wrong, since lipids of sample and SSM won't mix.
11. About co-reconstitution and normalization (Figure 4): excellent approach to tackle sensor-sensor-variations with different protein samples! Actually, the only way to really compare amplitudes across sensors when different transporters are used, which is important to compare Vmax values. This is in contrast to Figure 3, where normalization occurs with a reference substrate to only get information about relative substrate specificity for the different SMR homologues. Although, in terms of Vmax comparison with co-reconstitution, there might be still the factor of variable reconstitution efficiency, which cannot be tackled by this approach alone - did you check the protein concentration after reconstitution, i.e. via Western Blot? Just a minor thing, which would make the analysis just perfect. Still, with the internal reference alone it is a huge improvement over other rather rough comparisons of Vmax you'll find in literature so far, employing SSME.
12. "Although we initially sought to examine substrate binding by Gdx-pAmi as well, the protein requires high salt concentrations for purification and, in detergent, was prone to aggregate over long titrations or at more physiological salt concentrations." Great to actually read about experimental issues, very helpful for other researchers - this comes too short in many publications.
13. General comment regarding data in Figure 6: This kind of measurement could be also done using a pH dependence of the transport currents observed in SSME as a complementary approach, providing additional information, such as if H⁺ binding affects transport in a similar way as substrate binding. Check the following publication as reference: 10.1371/journal.pone.0156392 (Figures 2 and 3).
14. Page 13: "Major routes for the acquisition of new traits by a microbial population include the gain new genes via HGT transfer events,.." - there is something wrong with this sentence? Maybe "the gain of new genes"?

We thank the editor and reviewers for their constructive comments on our manuscript. We have updated our manuscript accordingly. Specific, point-by-point responses are detailed below:

Editor:

1. Please include an appropriate statistical measure to support your interpretation of differences. I expect that your P-values will be quite low, but you do need to calculate and display them.

We have updated figures 2 and 4 to include statistical tests of significance and p-values.

2. First, I think there is an error in equation 2, and that the numerator should be m, not n (i.e. m/(m-n)).

Thank you for catching that! We actually mis-identified n and m. (n is substrate and m is protons). The equation should be correct now:

$$E_{rev} = \left(\frac{n}{m - n} * \frac{RT}{F} \ln \frac{[substrate^+]_{in}}{[substrate^+]_{out}} \right)$$

where n and m represent the stoichiometric coefficients of substrate and protons, respectively

Second, and more important, an odd situation arises because you have no pH gradient, and because of the 2:1 stoichiometry of these proteins. In this case, m/(m-n) is -1, and so E_{rev} is equal to the Nernst potential for substrate (at least in absolute value), which would also be true if you had an uncoupled channel. In your case, though, since you're actually measuring the reversal of substrate-induced proton flux I think your data indeed support your conclusion-but I think its worth a few words of discussion to clarify for a reader who might be confused seeing Nernstian reversal for a transporter.

Ah, yes, that is slightly confusing. We have added the requested clarification to the text as follows (additions in italics).

With 10-fold higher external solute, the electrochemical equilibrium is expected to occur at -60 mV for coupled 2 H⁺: 1 solute transport. *(Note that this value has the inverse sign – and is thus far from – the Nernstian reversal potential for solute of +60 mV that would be expected for uncoupled solute flux).*

3. I think it would also be helpful to indicate on the reversal potential graph the expected values for alternate stoichiometries like 3:1 or 1:2

We added the following to the text: For a coupling ratio of 3 H⁺:1 solute, E_{rev} would be equal to -30 mV, and for a leaky transporter with a reduced coupling ratio of 1.7 H⁺:1 solute, E_{rev} would be -90 mV. We also added this information to the figure legend.

Reviewer #1

1. The authors attempt to use SSM electrophysiology to quantitatively compare transport rates of the four different SMRGdx homologues and two substrates. A quantitative comparison is usually impossible because of the fluctuation in current measurements between liposome prep, sensors, or even day-to-day experiments. The authors aim to circumvent this problem by co-reconstituting fluoride channel Fluc and using its currents as an internal standard. While I find this approach intriguing, I see two issues in the results shown in Figure 4.

The first issue is the absence of a non-activating solution in their solution exchange protocol (Figure 4A).

Our original experiments did indeed include perfusion with non-activating solutions between the fluoride and guanidinium pulses. We have clarified this in our description of the approach in the text and in the legend of Figure 4A. The text now reads:

“Control experiments with individually reconstituted Fluc-Bpe and SMR_{Gdx} confirm that the SMR_{Gdx} substrates guanidinium and guanylurea do not elicit a response from Fluc-Bpe, and that the SMR_{Gdx} are similarly insensitive to fluoride perfusion. *Between each substrate perfusion, we perfused with non-activating (substrate-free buffer) so that we could isolate the contribution of the Fluc or SMR_{Gdx} to the current.*”

2. Furthermore, if SSM is used to compare the differences in transport rates among the four homologues, it is essential to show consistent reconstitution efficiencies between all co-reconstituted proteins. If the authors did not intend for the homologues to be compared but just the two substrates, I suggest avoiding normalizing to a single homologue reference (Gdx-Clo). Otherwise, I propose including quantitative Western Blot analysis to assess the reconstitution efficiency of Fluc and all co-reconstituted proteins.

We performed the quantitative Western analysis as requested. This is shown in Figure 4E (full gels in Supplementary Figure 5), with quantification in Supplementary Table 2. Three of the SMR_{Gdx} homologues did not vary in reconstitution efficiency (statistically insignificant). However, one of the homologues, Gdx-pAmi exhibited a lower reconstitution efficiency than the others. Importantly, the reconstitution efficiency was not changed for any of the homologues by presence or absence of Fluc-Bpe.

We have adjusted the relative rate of Gdx-pAmi reported in Figure 4E to account for its reduced reconstitution efficiency, and added a section on the quantitative western blot to the methods section

3. The authors use a liposome-based pyranine assay to determine a substrate-to-proton stoichiometry of 1:2 for guanylurea and Gdm⁺. However, the results presented in Figure 5A need clarification. It is also unclear why the authors did not use the fluorescence ratio of pyranine by exciting at two instead of only one wavelength.

Exciting at two wavelengths allows for ratiometric measurement of the pH. We did not think this was necessary since we are not reporting precise pH values and are only interested in the change over time. We did ensure that the change in fluorescence was as expected when we excited at 400 nm rather than 455 nm.

How was the baseline fluorescence for the experiments defined by the authors? There appears to be an offset in fluorescence at baseline before the addition of guanylurea and valinomycin for the Gdx-Clo and Gdx-pAmi liposome samples.

For our experiments with guanylurea, we noted a small, reproducible decrease in the baseline upon guanylurea addition. We're not sure why this occurs – some interaction between the pyranine and guanylurea, perhaps? We are nonetheless confident in our conclusions about the stoichiometry, since the -60 mV traces shows no further change in fluorescence after this initial small jump, whereas both the -30 mV and -90 mV samples show notably larger changes in fluorescence over the timecourse that would be expected for a transport process.

Additionally, it would enhance the study to include control experiments with no substrate but valinomycin and vice versa.

We disagree that these experiments would be informative as controls, since both maneuvers change the electrochemical equilibrium. If substrate is added without valinomycin, coupled proton transport ensues

due to the substrate gradient (this is essentially just a fourth voltage, a “0 mV” measurement that would be expected to follow the same trend as the voltages already displayed). If valinomycin is added without substrate, the combination of a strong K^+ gradient and no other route for ions across the membrane drives K^+/H^+ antiport, which is not relevant to the measurements. Thus, these single addition experiments do not serve as effective negative controls. We note that this is a well-controlled experiment, with applied voltages both above and below the expected reversal potential, as well as using Gdx-Clo as a positive control (for which the stoichiometry has already been well-characterized).

Also, the authors should indicate the exact time point from the kinetic experiment shown in Figure 5A that was used for the graphs in Figure 5B. Including this information in Figure 5 would improve the clarity.

We have added this information to the legend:

“Measurements are a running 10 s. average of the final 10 s. of each trace (140-150 s. for Gdx-Clo with Gdm^+ and Gdx-pAmi with guanylurea, and 190-200 s. for Gdx-pAmi with guanylurea).”

3. If the authors could perform the ITC experiments in triplicates, this would considerably increase the robustness of their results. This becomes more important considering that, despite careful controls, fluctuations in the buffer composition could be the cause of the endothermic contribution reported during guanylurea titration.

We have updated Table 3 to report results (mean and SEM) from three independent experiments.

4. The authors should expand the ITC method section to include critical details such as titration volume, interval time, and differential power settings.

The following was added to the description of ITC in the methods section:

“For each experiment, 300 μ L of 700 μ M Gdx-Clo was loaded in the sample chamber maintained at 25°C with 350 rpm stirring speed. The injection syringe contained 500 μ L of buffer-matched substrate (20 mM Gdm^+ or 10 mM Guanylurea). Sample was titrated (0.75 μ L injections) at 100 second increments. In addition to the acceptable baseline absorbance slope (0.30 μ W/h and 0.03 μ W standard deviation), a 200 second baseline (~112 μ W) was taken prior to beginning titrations.”

5. Please provide more details for the SSM method section, such as the description of solution exchange protocols, primarily if they are not described in the figure legends.

We have added the following to the methods: *“For every SSM electrophysiology experiment, we used the same general solution exchange protocol: after 2 seconds perfusion with non-activating solution, we perfused substrate-containing buffer for 2 s, and then returned to the equilibrium condition with a 4 s perfusion of non-activating buffer.”*

We also included a new Supplementary Figure 1 showing an example experiment with all perfusions shown.

We also updated the appropriate figure legends to indicate that we are showing the substrate perfusion step only.

6. Tables 1 and 2 should contain legends to display how many technical or replicates they encompass.

We provided this information in the accompanying figures (Figures 3 and 6, respectively). We have now repeated this information in the table legends as well.

7. Figure 8 would benefit from also including the label "Side View" to more easily identify the different perspectives.

We have made the requested addition to this figure.

8. Provide a proper figure legend for the Supplementary Figure to offer the reader a clear understanding of the content in the main text.

We're not sure which legends required more information to interpret, but we have tried to provide the requested detail. (Added material shown in *Italics* below).

Supplementary Figure 1. Size exclusion chromatograms for six proteins in this study. *The major peak was collected for biochemical analysis.*

Supplementary Figure 2. Representative current traces for substrates and transporter data summarized in Figure 1, and no-protein controls. *Only the substrate perfusion step is shown.* Traces for a substrate series are from the same sensor. Box height is equal to current values shown at right.

Supp Fig. 3 Representative current traces for Gdm⁺ and guanylyurea titrations of protein-free liposomes. *Traces from a single sensor. Only the substrate perfusion step is shown.*

Supplementary Figure 4. Representative current traces for guanylyurea perfusion of Fluc-Bpe and fluoride perfusions of Gdx-Eco, Gdx-pPro, and Gdx-pAmi. *Only the substrate perfusion step is shown.*

Supp Fig. 6 Tryptophan fluorescence spectra and fits to binding isotherms for all data reported in Figure 6 and Table 2. *For the spectra of each representative titration series, the arrow indicates whether fluorescence increases or decreases with increasing substrate. Representative plots of substrate concentration versus change in fluorescence (from a single titration) are also shown and are fit to a single site binding isotherm (for Gdm⁺ titrations) or a single site binding isotherm with a linear correction for non-specific binding (for guanylyurea tirations) as described in the Methods section.*

Reviewer #2

The paper would significantly benefit from editorial work to produce a leaner version. Some paragraphs are repetitive and describe in unnecessary detail the rationale for experimental design (e.g., p.9: "Our experiments thus far do not..."; p. 11: "Although we initially sought..."; p.12: " Because tryptophan fluorescence; etc.). Is there a reason to go into so much detail describing methods previously published or writing the Nernst equation?

We've attempted to remove unnecessary text throughout. However, we also want to make sure it remains accessible to non-specialists, and we're hoping this finds an audience broader than mechanistic membrane transport people. A lot of microbiology transport people especially could probably benefit from seeing the Nernst equation written out! (Even if it is well known to all of us).

p.11: The authors state that separate control experiments showed that binding kinetics are fast and that the binding reaction achieved equilibrium before measurement. What is fast? Usually, binding per se is a very

fast reaction, not just fast. If this is not the case for the proteins studied, it may indicate some slow conformational changes required before binding. Is this the case?

We intended to convey that binding is fast compared to the conformational change to transport of the substrate. (This is not a surprising result, but we think it is important to explicitly test and report this, precisely so that unexpected pre-binding conformational changes do not confound our quantification.)

We have re-written this to say “separate control experiments showed that ~~binding kinetics are fast and that~~ the binding reaction achieved equilibrium before measurement.”

p. 10: Isn't stoichiometry for EmrE 2:1 also for most classical substrates? The way it is written sounds like varying stoichiometries is this transporter's signature.

We updated the sentence to include the modifier “some” transported substrates. We think we are being sufficiently circumspect by saying that deviations in stoichiometry have been “reported” and citing the literature.

“Prior studies have shown Gdx-Eco possesses a well-coupled 2 H⁺: 1 Gdm⁺ stoichiometry (Kermani et al., 2018; Thomas et al., 2021). However, for SMR_{Qac} EmrE, it has been reported that the transport stoichiometry differs among *some* transported substrates (Robinson et al., 2017). We therefore employed a proteoliposome assay to experimentally assess coupling stoichiometry of Gdx-Clo and plasmid-associated Gdx-pA_{mi}.”

How does the Glutamate pK compare with that of EmrE? I guess there were studies other than the NMR ones.

We have updated the text and cited the literature with the following addition:

“This value is in the same approximate range as the pK_a values of the central glutamates in other SMR homologues (Li et al., 2021; Morrison et al., 2015; Muth and Schuldiner, 2000).

Reviewer #3

1. In the methods section about SSME, "prior to fusion with DPhPC monolayer" => proteoliposomes do not fuse with the SSM, but they physically adsorb or attach.

Thank you. We have updated our nomenclature throughout the manuscript.

2. In the same section, what is meant by positive reference samples? Is a reference substrate used to activate the transporter within the same sensor after five measurement of different substrates to check stability? Maybe clarify a little here and if the "reference sample" means substrate, indicate the reference substrate as well.

The sentence was clarified as follows:

“For substrate screening experiments, reference substrate samples (Gdm⁺ for SMR_{Gdx}, and TPA⁺ for SMR_{Qac}) were checked periodically to test for stability of the sensor; if the current amplitude of the reference compound differed by more than 10% on one sensor, this indicated de-adsorption of liposomes, and the sensor was not used for further experiments.

3. Methods section, Pyranin Assay: "1mM Pyranine" - space is missing between 1 and mM; "~30 s. to" - replace dot by comma?

4. Methods section, Tryptophan Fluorescence: "1 μ M purified...", "5mM DM" - space missing;
5. Method section, structure: "Gdx-Clo (10mg/mL) and L10 monobody (10mg/mL," - space missing two times

Thanks, we have corrected these typos.

6. Minor concern about data visualization in Figure 2: "Datapoints represent at least three independent sensor preparations from at least two independent biochemical purifications." Am I understanding correctly, that the shown datapoints, averages and SEMs consider data from the same sensor, different sensors and different biochemical purifications at the same time? I don't feel well with this kind of representation/analysis, since usually datapoints from the same sensor show much lower variations compared to analyzing different sensors. And different biochemical purifications may show even higher variations in terms of amplitude. Typically I would only use one datapoint per sensor (or the average across all datapoints recorded within the same sensor) and subsequently average across sensors. Data from different biochemical purifications may be highlighted by color, but since the plot shows normalized data, I do not expect large differences between purifications beyond sensor-sensor variations.

We have updated the figure and the legend to better reflect the data reported (new text in italics). The figure is shown below. In our hands, the sensor-to-sensor variability obscures any prep-to-prep variability.

Figure 2. Screen for transport of metformin metabolites by SMR homologues. A. Chemical structures of metformin metabolites and metformin analog buformin. B. Amplitude of transport currents evoked by perfusion with 2 mM substrate. Current amplitudes are normalized to a positive control (Gdm⁺ for SMR_{Gdx} and TPA⁺ for SMR_{Qac}) collected on the same sensor. *Each datapoint represents a measurement from a single independent sensor. Sensors were prepared from at least two independent biochemical purifications; each biochemical preparation is represented by a different shaped point. The bars show the mean and SEM of measurements from different sensors.*

7. Page 9: "The current amplitudes reflect the initial rate of transport(Bazzone et al., 2017), and their concentration dependence follows Michaelis-Menten kinetics (Figure 3, Table 1)." - this is true, providing a certain line of evidence, i.e. absence of pre steady-state currents. Two more recent publications from 2023 focus on this topic, which would be better citations: 10.1016/j.bios.2021.113763, 10.3389/fphys.2023.1058583 (the second one actually highlights and compares analysis methods to distinguish transport from binding events recorded with SSME).

We have added the citations suggested. We actually do have evidence from crosslinked proteins that the currents represent transport and not pre-steady state binding (at least for Gdx-Clo). Prompted by your question, we decided to include this data in this manuscript. This can now be found in Figure 2C-2F, with the following added to the text:

"For all SSM electrophysiology experiments, the shapes of the substrate-induced currents are characteristic of transport, rather than electrogenic pre-steady binding events. However, we sought to confirm this interpretation for at least one transporter/substrate pair. Using the structure of Gdx-Clo(Kermani et al., 2020), we introduced a pair of cysteines, A70C and S104C, that are in close proximity on the open side of the transporter, but that increase in distance when the transporter changes conformation. We expected that the introduced cysteine residues would crosslink and lock the transporter in one conformation, impairing transport, but that the mutations would have a limited effect on substrate binding (Figure 2C). Indeed, under oxidizing conditions, the K_d for Gdm⁺ binding is within a factor of two of WT (Nelson et al., 2017), but radioactive Gdm⁺ exchange is greatly reduced to near-background levels (Figure 2D, 2E). Transport is restored in reducing conditions. SSM electrophysiology recapitulates

this observation: under oxidizing conditions, the SSM electrophysiology traces of Gdx-Clo A70C/S104C are indistinguishable from those of protein-free liposomes, but inclusion of a reducing agent elicits characteristic transport currents (Figure 2F). Although Figure 2D shows that Gdx-Clo A70C/S104C binds substrate normally when locked, we do not see any evidence of pre-steady state binding currents."

9. Later on page 9: "We verified that protein-free liposomes did not exhibit currents at all substrate concentrations tested (Supplementary Figure 3)." - there are small currents at the highest concentrations used. I would be more defensive in this sentence, like "only the highest substrate concentrations show small currents, in the range of <x % compared to the protein sample..."

We updated the sentence to read:

"Protein-free liposomes do not exhibit negative capacitive currents characteristic of transport; at the highest substrate concentrations we observe small positive currents, indicative of interactions with the membrane (Supplementary Figure 3).

10. Last paragraph, page 9: "As with other measurements that rely on liposome fusion to a lipid bilayer(Stockbridge and Tsai, 2015),..." - SSME does not rely on the fusion, but adsorption as pointed out later in the same sentence. However, later "fusion efficiency" is mentioned. I would consistently say "adsorption", if possible. Fusion is wrong, since lipids of sample and SSM won't mix.

We appreciate the clarification. As noted above, we changed this throughout the manuscript.

11. About co-reconstitution and normalization (Figure 4): excellent approach to tackle sensor-sensor-variations with different protein samples! Actually, the only way to really compare amplitudes across sensors when different transporters are used, which is important to compare Vmax values. This is in contrast to Figure 3, where normalization occurs with a reference substrate to only get information about relative substrate specificity for the different SMR homologues. Although, in terms of Vmax comparison with co-reconstitution, there might be still the factor of variable reconstitution efficiency, which cannot be tackled by this approach alone - did you check the protein concentration after reconstitution, i.e. via Western Blot? Just a minor thing, which would make the analysis just perfect. Still, with the internal reference alone it is a huge improvement over other rather rough comparisons of Vmax you'll find in literature so far, employing SSME.

Thanks for the positive comments about this approach. We now include Western blot analysis in Figure 4 as requested (see response to reviewer #1).

12. "Although we initially sought to examine substrate binding by Gdx-pAmi as well, the protein requires high salt concentrations for purification and, in detergent, was prone to aggregate over long titrations or at more physiological salt concentrations." Great to actually read about experimental issues, very helpful for other researchers - this comes too short in many publications.

Thank you!

13. General comment regarding data in Figure 6: This kind of measurement could be also done using a pH dependence of the transport currents observed in SSME as a complementary approach, providing additional information, such as if H⁺ binding affects transport in a similar way as substrate binding. Check the following publication as reference: 10.1371/journal.pone.0156392 (Figures 2 and 3).

Thanks for the suggestion. We'll do this in the future.

14. Page 13: "Major routes for the acquisition of new traits by a microbial population include the gain

new genes via HGT transfer events,.. " - there is something wrong with this sentence? Maybe "the gain of new genes"?

Thanks! Fixed it.

December 21, 2023

Prof. Randy B. Stockbridge
University of Michigan-Ann Arbor
MCDB and Biophysics
1105 N. University Ave.
Biological Sciences Building Room 3242
Ann Arbor, MI 48109

Re: 202313464R1

Dear Prof. Stockbridge,

I am pleased to let you know that your manuscript, entitled "Transport of metformin metabolites by guanidinium exporters of the Small Multidrug Resistance family" is scientifically acceptable for publication in Journal of General Physiology. Formal acceptance will follow when it is modified in accordance with the referees' remarks and our editorial policies.

Please note items that need attention are listed at the bottom of this email (under 'manuscript formatting checklist') and on the attached marked-up pdf file. Please also be sure to include a letter addressing the reviewers' comments point-by-point (if applicable) and a copy of the text with alterations highlighted (boldfaced or underlined). Your manuscript should be a double-spaced MS Word file and include editable tables, if appropriate.

JGP now requires a data availability statement for all research article submissions. These statements will be published in the article directly above the Acknowledgments. The statement should address all data underlying the research presented in the manuscript. Please visit the JGP instructions for authors for guidelines and examples of statements at <https://rupress.org/jgp/pages/editorial-policies#data-availability-statement>.

Lastly, JGP adds short captions to articles listed on our weekly newest article emails. If you haven't, please provide a short, ~40-word summary statement for the online JGP table of contents and alerts. This summary should describe the context and significance of the findings for a general readership and be placed on/near the title page.

Please submit your final files via this link:
Link Not Available

Thank you for choosing to publish your research in JGP and please feel free to contact me with any questions.

Sincerely,

Joseph A. Mindell, M.D., Ph.D.
On behalf of Journal of General Physiology

Journal of General Physiology's mission is to publish mechanistic and quantitative molecular and cellular physiology of the highest quality; to provide a best in class author experience; and to nurture future generations of independent researchers.

Manuscript formatting checklist:

- MS Word document of text needed (including editable tables)
- MS Word document of supplemental text needed, if applicable (including figure legends and editable tables)
- Brief Statement describing supplementary information needed, if applicable (in subsection at end of Materials & Methods)
- Please include a data availability statement preceding the Acknowledgments section. Please see <https://rupress.org/jgp/pages/editorial-policies#data-availability-statement>
- Figures created at sufficient resolution and in acceptable format (including supplemental if applicable). If working in Illustrator, we prefer .ai or .eps file format. If working in Photoshop please use 600dpi/1000dpi .tiff or .psd file format. Minimum resolution at estimated print size: Minimum resolution for all figures is 600 dpi. For figures that contain both photographs and line art or text, 600 dpi is highly recommended. Figures containing only black and white elements (line art, no color, and no gray) should be 1,000 dpi. Maximum figure size is 7 in wide x 9 in high (17.5 x 22.8 cm) at the correct resolution. <https://jgp.rupress.org/fig-vid-guidelines>
- Supplemental figures, if any, conforming to same guidelines as manuscript figures (noted above)
- If images resemble one from a prior publications, the author must seek permissions (to reproduce or adapt) from the original publisher. [You can resubmit your paper while waiting to hear back from the original publisher but please keep us updated]

- All authors must complete a disclosure form prior to acceptance. A link to complete the form has been sent to all coauthors. Please provide the editorial office with updated email addresses if necessary

Reviewer #1 (Comments to the Authors):

I appreciate the authors' detailed reply to the review and thank them for addressing all my comments. The additional experiments and the changes to figures and methods significantly improved their work. The crosslink experiments are a beautiful addition supporting their data. Thank you for including them!

Below is the summary of the points addressed by the authors and my reply to them.

1. The authors addressed my concerns over the SSM solution exchange protocol and included a quantitative comparison.

I thank the authors for providing details on the method and supplemental sections on the SSM solution exchange protocol. I also appreciate the inclusion of the quantitative Western analysis.

However, I am slightly concerned about the quantitative western blot used to normalize the relative current of Gdx-pAmi to Gdx-Clo because overexposure was required for one and not the other. I still agree with the authors' conclusion about seeing no difference in Gdx-pAmi relative transport rates between Gdm+ and guanilylurea.

2. In response to the pyranine assay methodology and suggested control experiments.

I thank the authors for their explanations and understand the decision not to use two wavelengths for pH measurement. I also agree with the rationale behind excluding additional controls. Also, thank you for sharing your observation on the small baseline decrease after adding guanilylurea.

3. In response to the ITC experiments.

I appreciate the effort in performing additional ITC experiments and the changes made to the methods section.

4. The authors have responded to various suggestions, including adding details to the SSM method section, providing table legends, adding labeling to Figure 8, and enhancing figure legends for supplementary figures.

Thank you for taking the time to implement all these changes.

Some very minor revisions:

-Line 354/5 has a sentence that needs to be fixed.

-In the figure legend of Supplementary Figure 6, you might want to change the word "Gel..." to membrane since this is a Western blot.

-Please include crosslinking conditions in the SSM and tryptophan fluorescence method sections.

Reviewer #2 (Comments to the Authors):

Accept.

Reviewer #3 (Comments to the Authors):

all my concerns and suggestions have been addressed.